



# Profiling of formaldehyde, glyoxal, methylglyoxal, and CO over the Amazon: Normalised excess mixing ratios and related emission factors in biomass burning plumes

Flora Kluge[1], Tilman Hüneke[1,i], Matthias Knecht[1,ii], Michael Lichtenstern[2], Meike Rotermund[1], Hans Schlager[2], Benjamin Schreiner[1], and Klaus Pfeilsticker[1]

[1]Institute of Environmental Physics, University of Heidelberg, Heidelberg, Germany
[2]Institut für Physik der Atmosphäre, Deutsches Zentrum für Luft- und Raumfahrt (DLR), Oberpfaffenhofen, Germany
[i]now with Encavis AG, Hamburg, Germany
[ii]now with Ernst & Young, GmbH, Wirtschaftsprüfungsgesellschaft, Stuttgart, Germany

**Correspondence:** Flora Kluge
(fkluge@iup.uni-heidelberg.de)

**Abstract.** We report on airborne measurements of tropospheric mixing ratios and vertical profiles of $CH_2O$, $C_2H_2O_2$, $C_3H_4O_2^*$, and CO over the Amazon Basin during the ACRIDICON-CHUVA campaign from the German High Altitude and Long-range research aircraft (HALO) in fall 2014. The joint observation of in situ CO and remotely measured $CH_2O$, $C_2H_2O_2$, and $C_3H_4O_2^*$, together with visible imagery and air mass back trajectory modelling using NOAA HYSPLIT (National Oceanic

5   Atmospheric Administration, HYbrid Single-Particle Lagrangian Integrated Trajectory) allow us to discriminate between the probing of background tropical air, in which the concentration of the measured species results from the oxidation of biogenically emitted VOCs (mostly isoprene), and measurements of moderately to strongly polluted air masses affected by biomass burning emissions or the city plume of Manaus. For twelve near surface measurements of fresh biomass burning plumes, normalised excess mixing ratios of $C_2H_2O_2$ and $C_3H_4O_2^*$ with respect to $CH_2O$ are inferred and compared to recent studies. The

10   mean $R_{GF}$=0.07 (range 0.02–0.11) is in good agreement with recent reports which suggest $R_{GF}$ to be significantly lower than previously assumed in global CTM models. The mean $R_{MF}$=0.98 (range 0.09–1.50) varies significantly during the different observational settings, but overall appears to be much larger (up to a factor of 5) than previous reports suggest even when applaying a correction factor of $2.0 \pm 0.5$ to account for the additional dicarbonyls included in the $C_3H_4O_2^*$ measurements. Using recently reported emission factors of $CH_2O$ for tropical forests, our observations suggest emission factors of $EF_G$=0.25

15   (range 0.11 to 0.52) $g\,kg^{-1}$ for $C_2H_2O_2$, and $EF_M = 4.7$ (range 0.5 to 8.64) $g\,kg^{-1}$ for $C_3H_4O_2^*$. While $EF_G$ agrees well with recent reports, $EF_M$ is (like $R_{MF}$) slightly larger than reported in other studies, presumably due to the different plume ages or fuels studied. Our observations of these critical carbonyls and intermediate oxidation products may support future photochemical modelling of air pollution over tropical vegetation, as well as validate past and present space-borne observations of the respective species.





## 1 Introduction

Emissions from biomass burning are a large source of reactive gases and particulate matter to the atmosphere on a local, regional and up to the global scale (Crutzen and Andreae, 1990; Andreae and Merlet, 2001; Andreae et al., 2001; Andreae, 2019). Pyrogenic emissions form a complex mixture of species that can change over the duration of the fire and undergo further chemical reactions as emissions are transported downwind. In consequence, biomass burning may have adverse effects on the atmospheric photochemistry, secondary aerosol formation and composition, and hence cloud formation and radiation, as well as human health. Therefore, accurate measurements of pyrogenic emissions, the evolution of fire plume composition and the formation and aging of aerosols are needed, but respective data are lacking for many species (e.g., Andreae and Merlet (2001); Akagi et al. (2011); Stockwell et al. (2015); Andreae (2019)).

Among the manifold of species emitted in large amounts by fires are carbonyl compounds such as formaldehyde ($CH_2O$), glyoxal ($C_2H_2O_2$), methylglyoxal ($C_3H_4O_2$), 2,3-butanedione ($C_3H_6O_2$), and many others (e.g., Andreae (2019)). Here and further on, $C_3H_4O_2^*$ is denotes $C_3H_4O_2$ and other substituted dicarbonyls (cf. 2,3-butanedione) with visible absorption spectra similar to $C_3H_4O_2$, since they cannot be distinguished with the spectral resolution available in our limb measurements (sect. 2.1).

It is well known that in the atmosphere most of $CH_2O$ is formed as an intermediate oxidation species from the degradation of $CH_4$ and from a suite of non-methane hydrocarbons (NMHCs) (Seinfeld and Pandis, 2013). $CH_2O$ can further be directly emitted into the troposphere by biomass burning (e.g., Finlayson-Pitts and Pitts (2000); Andreae and Merlet (2001); Wagner et al. (2002); Akagi et al. (2011); Seinfeld and Pandis (2013); Stockwell et al. (2015)), and from incomplete combustion as well as the direct emission by vegetation (Carlier et al. (1986) and references therein). Due to its short lifetime in the atmosphere of only several hours, $CH_2O$ mixing ratios range from several 10 ppt to a few 100 ppt in the pristine (e.g., marine) environment. In the lower or polluted troposphere, mixing ratios might reach several ppb (e.g., Finlayson-Pitts and Pitts (2000); Jaeglé et al. (1997); Wagner et al. (2002); Borbon et al. (2012); Seinfeld and Pandis (2013) and others). Secondary formed $CH_2O$ (for example from acetone photolysis) is an important source of $HO_x$ radicals, in particular in the middle and upper troposphere (Jaeglé et al., 1997). Further, since $CH_2O$ is considerably water soluble, it may participate in either acid or base catalysed aldol-condensation reactions in the aerosol phase, hence contributing to secondary organic aerosol (SOA) formation (Wang et al., 2010). Numerous studies of formaldehyde emissions from biomass burning both in the laboratory and the field were undertaken in the past, and its emission factor appears to be fairly well established for the various studied fuel and fire types (e.g., Finlayson-Pitts and Pitts (2000); Andreae and Merlet (2001); Wagner et al. (2002); Akagi et al. (2011); Seinfeld and Pandis (2013); Stockwell et al. (2015); Andreae (2019)). Based on this knowledge, models have generally been able to reproduce reasonably well the formaldehyde column densities observed by satellites (e.g., Dufour et al. (2009); Stavrakou et al. (2009); Boeke et al. (2011); Bauwens et al. (2016), and others), or concentrations directly measured in the atmosphere (e.g., Arlander et al. (1995); Fried et al. (2008); Borbon et al. (2012); Kaiser et al. (2015); Chan Miller et al. (2017) and others).

Similar to formaldehyde, glyoxal and methylglyoxal are formed by 47 % and 79 %, respectively, during the oxidation of isoprene emitted by vegetation (Fu et al. (2008); Wennberg et al. (2018)). Recent air-borne and satellite measurements





identified the Amazon to be the world most prominant isoprene source region, with mixing ratios exceeding 6 ppb isoprene in the boundary layer by end of the dry season in September 2014 (Gu et al., 2017; Fu et al., 2019). These elevated mixing ratios are a consequence of the large isoprene emission from the rain forest combined with a significantly prolonged atmospheric lifetime of up to 36 hours (GEOS-Chem model simulations) over the Amazon Basin compared to the global mean of less than 4 hours (Fu et al. (2019), fig. 6a). Fu et al. (2019) suspected strong isoprene emissions combined with low $NO_x$ concentrations to lead to significantly supressed OH-levels as possible reason for the observed slower isoprene oxidation in this regime. The longer isoprene lifetime and efficient vertical transport over the Amazon lead to elevated isoprene mixing ratios not only within the boundary layer but also in higher altitudes (Fu et al. (2019), fig. 4d), where it is a possible source for in situ glyoxal and methylglyoxal production in the free troposphere.

The second most important precursors of the trace gases are acetylene (mostly anthropogenicly emitted) for glyoxal and acetone (mostly biogenic) for methylglyoxal, but both gases are also formed during the oxidation of other volatile organic compounds (VOCs), particularly alkenes, aromatics, and monoterpenes (Fu et al., 2008). For both gases, Fu et al. (2008) estimated their global source strengths to 45 $Tg\,a^{-1}$ and 140 $Tg\,a^{-1}$, respectively. Atmospheric lifetimes of glyoxal and methylglyoxal are only about 2 h, mostly due to photolysis and to a lesser degree due to reactions with OH radicals (Koch and Moortgat, 1998; Volkamer et al., 2005a; Tadić et al., 2006; Fu et al., 2008; Wennberg et al., 2018). Accordingly, in ambient air $C_2H_2O_2$ and $C_3H_4O_2$ mixing ratios typically do not exceed several 100 ppt (Fu et al., 2008).

Glyoxal and methylglyoxal are additionally emitted in considerable quantities by biomass burning, their individual emission strengths however are not yet well established (e.g., Andreae and Merlet (2001); Akagi et al. (2011); Stockwell et al. (2015); Zarzana et al. (2017, 2018); Andreae (2019)). The recent compilation of emission factors by Andreae (2019) indicates for most reported fuels the complete lack of emission data for both dicarbonyls. Until the recent laboratory and field studies by Zarzana et al. (2017, 2018), previous information on the glyoxal and methylglyoxal emissions from biomass burning were based on only two laboratory studies (McDonald et al., 2000a; Hays et al., 2002). While (Hays et al., 2002) reported larger emission factors for glyoxal compared to methylglyoxal, the recent findings of Zarzana et al. (2017, 2018) indicated significantly smaller glyoxal emissions but larger emissions of methylglyoxal and other substituted dicarbonyls in fires of agricultural biomass than those previously assumed in models (Fu et al., 2008).

Satellite and aircraft measurements of formaldehyde and glyoxal complemented by modelling have been used to study their different atmospheric sources (e.g., Stavrakou et al. (2009); Boeke et al. (2011); Lerot et al. (2010); Chan Miller et al. (2014); Bauwens et al. (2016); Stavrakou et al. (2016), and others), the secondary aerosol formation from carbonyls (typically in the background atmosphere), as well as in biomass burning plumes (Knote et al., 2014; Li et al., 2016; Lim et al., 2019), and even more recently to estimate the organic aerosol abundance (Liao et al., 2019). Unlike for formaldehyde, the models had varying success in reproducing the glyoxal column densities (Myriokefalitakis et al., 2008; Stavrakou et al., 2009; Lerot et al., 2010). In comparison to satellite measurements from SCIAMACHY and GOME-2, several studies have found that the models underestimate global glyoxal emissions, when not considering additional secondary sources apart from those inferred from the oxidation of biogenic precursors.





The ratios of $C_2H_2O_2$ to $CH_2O$ ($R_{GF}$), and $C_3H_4O_2$ to $CH_2O$ ($R_{MF}$) can be used as indications of the precursor VOC species and to identify biomass burning in contrast to biogenic sources (Kaiser et al., 2015). In the past, $R_{GF}$ has been studied based on multiple ground-based observations as well as satellite measurements (e.g., McDonald et al. (2000a); Kaiser et al. (2015); Chan Miller et al. (2017)). Kaiser et al. (2015) studied inferred $R_{GF}$ over the south-eastern United States during the 2013 SENEX (Southeast Nexus) campaign, and compared these measurements to respective OMI (Ozone Monitoring Instrument) satellite retrievals. They concluded, that in regimes rich of monoterpene emissions $R_{GF}$ is generally larger than $3\%$, while regions dominated by isoprene oxidation appear characterized by low $R_{GF} < 3\%$. For a compilation of previously inferred $R_{GF}$ see Kaiser et al. (2015), Table 1.

More recently, Stavrakou et al. (2016) examined emissions of crop residue fires in the North China Plain using data from the OMI satellite, which is part of the so called A-train (https://atrain.nasa.gov/). Their inferred column emission ratio of 0.04–0.05 was comparable to the $R_{GF}$ observed by Zarzana et al. (2017), but smaller than those reported by McDonald et al. (2000a) and Hays et al. (2002), which were used in the study of Fu et al. (2008). Nevertheless, the chemical transport model used by Stavrakou et al. (2016) was able to reproduce the measured formaldehyde column densities and the glyoxal enhancements observed over the North China Plain during the peak burning season.

Information on methylglyoxal in the atmosphere is even more rare, since to our knowledge satellite instruments measuring methylglyoxal are not (yet) available, and atmospheric measurements of methylglyoxal are still sparse. This is a result of the moderate spectral resolution of space- and air-borne optical measurements, which suffer from the spectral interference of the major but only weakly structured absorption bands of methylglyoxal with those of other dicarbonyls (mainly 2,3-butanedione) in the blue spectral region (Meller et al., 1991; Horowitz et al., 2001; Thalman et al., 2015; Zarzana et al., 2017). Potential interferences with the $7\nu$ absorption band of water vapour at $442\,nm$ further complicate the spectral retrieval (Thalman et al., 2015; Zarzana et al., 2017). In situ measurements of methylglyoxal using different techniques have only recently become available (Thalman and Volkamer, 2010; Pang et al., 2011; Lawson et al., 2015; Michoud et al., 2018). The very limited number of measurements in conjunction with insufficient detection limits are still preventing comprehensive studies of methylglyoxal.

Here we report on simultaneous measurements of $CO$, $CH_2O$, $C_2H_2O_2$, and $C_3H_4O_2^*$ in the lower to upper troposphere over the Amazon Basin at the end of the dry season in 2014. All measurements were performed from the german research aircraft HALO operated by the Deutsches Zentrum für Luft- und Raumfahrt (DLR). The simultaneous near surface measurements of $C_2H_2O_2$ and $C_3H_4O_2^*$ allow us to estimate their emission ratios with respect to $CH_2O$ as well as the normalized excess mixing ratios within the probed biomass burning plumes, and when combined with previously reported emission factors (e.g., Andreae (2019)), to estimate their emission factors for tropical forest fires. The joint measurements of the carbonyls and $CO$ in the middle and in the upper troposphere, more distant from their direct sources at the ground, provides insight into their tropospheric profiles and precursors in a pristine atmosphere and under low $NO_x$ conditions. Additionally, air mass backward trajectory modelling classifies the probed air masses and their possible origins.

The paper is organized as follows. Chapter 2 briefly describes the measurement techniques and methods used in the present study. Chapter 3 and 4 report on the ACRIDICON-CHUVA campaign and the actual measurements, and chapter 5 discusses the major findings and results. Chapter 6 concludes and summarises the study.





**Table 1.** Trace gas absorption cross sections used for the DOAS based spectral retrieval.

| No. | Absorber | Temperatur [K] | Reference | Uncertainty [%] |
|-----|----------|----------------|-----------|-----------------|
| 1 | $O_4$ | 273 | Thalman and Volkamer (2013) | 4 |
| 2 | $O_3$ | 203 | Serdyuchenko et al. (2014) | 3 |
| 3 | $O_3$ | 273 | Serdyuchenko et al. (2014) | 3 |
| 4 | $NO_2$ | 273 | Burrows et al. (1998) | 4 |
| 5 | $H_2O$ | 293 | Rothman et al. (2009) | |
| 6 | $CH_2O$ | 293 | Chance and Orphal (2011) | 10 |
| 7 | $C_2H_2O_2$ | 296 | Volkamer et al. (2005b) | 3 |
| 8 | $C_3H_4O_2$ | 296 | Meller et al. (1991) | 10 |
| 9 | $C_4H_6O_2$ | 223 | Horowitz et al. (2001) | 4 |

## 2 Methods

### 2.1 DOAS Measurements

The mini-DOAS instrument is a UV/vis/near-IR six channel optical spectrometer, which detects scattered skylight received from scanning limb and nadir directions. These spectra can be analysed for absorption structures of $O_4$, $CH_2O$, $C_2H_2O_2$ and $C_3H_4O_2^*$ (among other species). A video camera (type uEye UI-1005XS from IDS Imaging Development Systems GmbH) with a field of view of 46° is installed into the telescope unit and co-aligned with the spectrometer's limb telescopes. It provides visual imagery of the probed atmospheric scene at a rate of 1 Hz. An extensive description of the instrument and its deployment

on the HALO aircraft can be found in Hüneke (2016); Hüneke et al. (2017). The mini-DOAS instrument remotely probes the atmosphere with the limb telescopes (FOV 0.5° vertical, 3.15° horizontal) oriented perpendicular and right hand to the aircraft main axis. Although the limb telescopes can be directed at any elevation angle for traditional atmospheric profiling, during the ACRIDICON-CHUVA campaign they were constantly directed at 0.3° below the horizon, for optimal measurements when using the $O_4$ scaling method (Hüneke (2016); Hüneke et al. (2017); Stutz et al. (2017); Werner et al. (2017), and below).

The post-flight analysis of the collected limb spectra includes a DOAS (Differential Optical Absorption Spectroscopy (Platt and Stutz, 2008)) analysis of the measured skylight spectra for the absorption structures of $O_4$, $CH_2O$, $C_2H_2O_2$, and $C_3H_4O_2^*$ as well as the attribution of the measured absorption to the correct locations in the atmosphere. The latter is performed using the $O_4$ scaling method described in detail in Hüneke (2016); Hüneke et al. (2017); Stutz et al. (2017) as well as below. The DOAS analysis of the data is performed according to the settings in Tables 1 and 2.

Exemplary sample retrievals of $CH_2O$, $C_2H_2O_2$, and $C_3H_4O_2^*$ are shown in fig. 1. The detection of $C_3H_4O_2^*$ (fig. 1, panel (3)) deserves a brief discussion. It is known that substituted dicarbonyls such as biacetyl ($CH_3COCOCH_3$), with absorption cross sections in the visible spectral range similar to those of $C_3H_4O_2$, are also co-emitted from biomass fires in significant quantities (e.g., Meller et al. (1991); Horowitz et al. (2001); Thalman et al. (2015); Zarzana et al. (2017)). Owing to the mod-




**Table 2.** Details of the DOAS spectral analysis for the various trace gases.

| Target gas | Spectral interval [nm] | Fitted absorbers (see Table 1) | Add. Param. | Polyn. | $\sigma$ [dSCD] |
|---|---|---|---|---|---|
| O$_4$ | 348–369 | 1, 2, 3, 4, 6 | $I_{Ofs}^{(i)}$, $R^{(ii)}$, $R \cdot \lambda^{4(iii)}$ | 2 | $8 \times 10^{41}$ |
| | 460–490 | 1, 2, 3, 4, 5 | $I_{Ofs}$, $R$, $R \cdot \lambda^4$ | 2 | $4 \times 10^{41}$ |
| CH$_2$O | 324–354 | 1, 2, 3, 4, 6 | $I_{Ofs}$, $R$, $R \cdot \lambda^4$ | 2 | $5 \times 10^{15}$ |
| C$_2$H$_2$O$_2$ | 420–439 and 447–465 | 1, 3, 4, 5, 7 | $I_{Ofs}$, $R$, $R \cdot \lambda^4$ | 2 | $1 \times 10^{15}$ |
| C$_3$H$_4$O$_2^*$ | 420–475 | 1, 3, 4, 5, 7, 8, 9 | $I_{Ofs}$, $R$, $R \cdot \lambda^4$ | 2 | $1 \times 10^{16}$ |

(i) $I_{Ofs}$: Offset spectrum; (ii) $R$: Ring spectrum; (iii) $R \cdot \lambda^4$: Ring spectrum multiplied by $\lambda^4$.

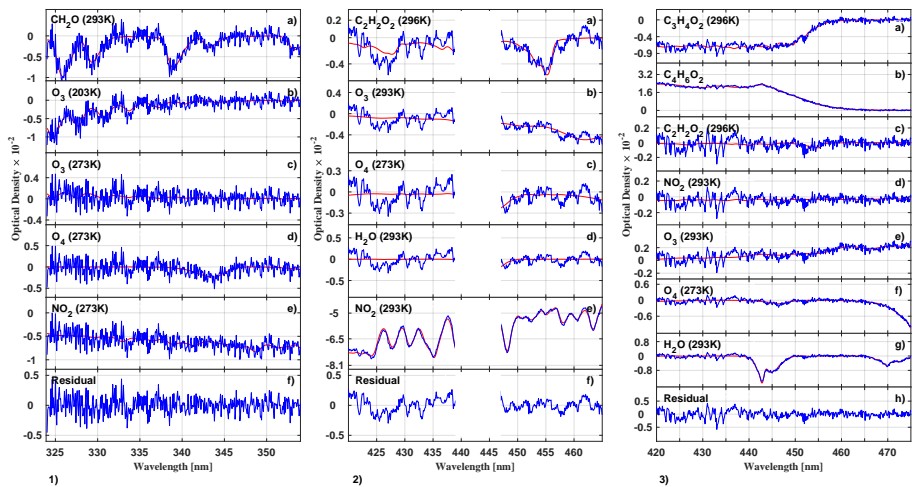

**Figure 1.** Inferred absorption spectra of CH$_2$O (1), C$_2$H$_2$O$_2$ (2), and C$_3$H$_4$O$_2^*$ (3) for the measurements at 17:16, 15:06 and 14:53 UTC, respectively, during the flight on Sept. 11, 2014 (AC11). The traces shown in blue are the inferred atmospheric spectra together with the residual spectral structures, the red line shows the reference spectra of the respective gases. In order to avoid cross correlations with the $7\nu$ absorption band of H$_2$O at 442 nm, C$_2$H$_2$O$_2$ is simultaneously fit over two separated spectral windows, ranging from 420 nm to 439 nm and 447 nm to 465 nm.

erate resolution of our spectrometer in the visible spectral range (full width half maximum resolution of 1.1 nm or 8.4 pixels, see Hüneke (2016)), we cannot separately (and thus unambiguously) detect these substituted dicarbonyls in the atmosphere, unlike in studies employing higher resolving spectrometers (e.g., Thalman et al. (2015)). Accordingly, we report here the weighted sum (absorption cross section times concentration) of C$_3$H$_4$O$_2$ and that of other substituted dicarbonyls and express that quantity as C$_3$H$_4$O$_2^*$.

From the retrieved slant column densities, absolute concentrations of the targeted gases are inferred using the recently developed O$_4$ scaling method. As shown in previous studies, the scaling method is based on simultaneous measurements of the target gas X and a scaling gas P of known concentration (here the collision complex O$_2$−O$_2$, briefly called O$_4$) in the same





wavelength interval (Hüneke, 2016; Hüneke et al., 2017; Stutz et al., 2017; Werner et al., 2017). The concentration $[X_j]$ of the trace gas X in the atmospheric layer j is then determined from

$$[X]_j = \frac{\alpha_{X_j}}{\alpha_{P_j}} \cdot \frac{SCD_X}{SCD_P} \cdot [P]_j \tag{1}$$

(see Stutz et al. (2017), eq. (14)). Here, $[P]_j$ is the calculated extinction of $O_4$ ($\epsilon_{O_4}=\sigma \cdot k_{eq} \cdot [O_2]_j^2$) in the atmospheric layer j, from which we take the sample. $SCD_P$ is the measured optical depth of $O_4$ and $SCD_X$ is the measured slant column density of the target gas X. The $\alpha$-factors $\alpha_{X_j}$ and $\alpha_{P_j}$ each express the ratio of the absorption of the gas in layer j to the total atmospheric absorption. Usually, the $\alpha$-factors are simulated in supporting radiative transfer calculations using the Monte Carlo model McArtim (Deutschmann et al., 2011; Knecht, 2015).

When applying the scaling method to air-borne UV/vis spectroscopy in limb geometry, a powerful radiative transfer model is necessary to treat the radiative transfer in 2D or 3D, calculate the refraction and also account for the sphericity of the Earth. We use the Monte Carlo radiative model McArtim (Deutschmann et al., 2011), where each measurement is modelled forward while considering the actual atmospheric, instrumental, and observational (e.g., Sun position relative to the pointing) parameters, the geolocation and the pointing of the telescope as described in Hüneke (2016); Hüneke et al. (2017). Further, a

combination of climatological aerosol profiles obtained by LIVAS (Amiridis et al., 2015) and the Stratospheric Aerosol and Gas Experiment II (SAGE II) (https://eosweb.larc.nasa.gov/project/sage3/sage3_table) is considered. For the calculation of the $\alpha$-factors, box air mass factors are simulated and summed, weighted by concentration (Stutz et al. (2017), eq. (11)). We calculate the extinction $[P]_j$ to obtain the vertical $O_4$ profile. For the target gases $CH_2O$, $C_2H_2O_2$, $C_3H_4O_2^*$, we assume exponentially decaying profiles. The latter appears justified because of photochemical arguments, the source distribution (mostly surface), the

photochemical lifetimes, the time scales of possible vertical transport, as well as when inspecting the profile shapes obtained in the post-analysis (fig. 9).

Evidently, UV/vis limb measurements naturally divide the atmosphere into three different layers, which are separated by the different contributions to the total absorption or optical depth (OD). These contributions are (a) the overhead absorption ($OD_{oh}$), (b) the absorption located within the line-of-sight of the telescopes ($OD_{limb}$) (of which most, but not all is due to

single scattering), and (c) the absorption below the line of sight ($OD_{ms}$) (of which the photon paths are necessarily all of higher scattering order). For a high-flying aircraft, $OD_{oh}$ has the least uncertainty since $O_4$ absorption approximately scales with $[O_2]^2$ and the optical state of the overhead atmosphere is well known. Therefore, the reminder of the measured total $O_4$ absorption has to be due to the varying contributions of (b) and (c), which are mostly modulated by the current low level cloud cover and ground albedo (Stutz et al. (2017), fig. 7). The knowledge of the $\alpha$-factor however gives a handle on the relative fractions (b) and

(c), since contribution (b) can be calculated from $OD_{limb}=\alpha \cdot OD_{meas}$, and accordingly $OD_{ms}=OD_{meas}-OD_{oh}-OD_{limb}$. Still, the calculated attributions of (b) and (c) to $OD_{meas}$ can only be approximated due to the assumptions made regarding the current aerosols distribution and the cloud coverage. Figure 2 illustrates the contributions to $OD_{meas}$ at 343.7 nm (panel (1a)) and 477.3 nm (panel (2a)) as a function of the flight time for Sept. 16, 2014. Figure 3 shows the same data, but sorted by flight altitude to demonstrate the relative change of the contributions. For both investigated wavelengths, the absorption within the

line of sight of the telescopes to the total $O_4$ absorption dominates with $> 50\%$ up to 10 km altitude, with a relative minimum

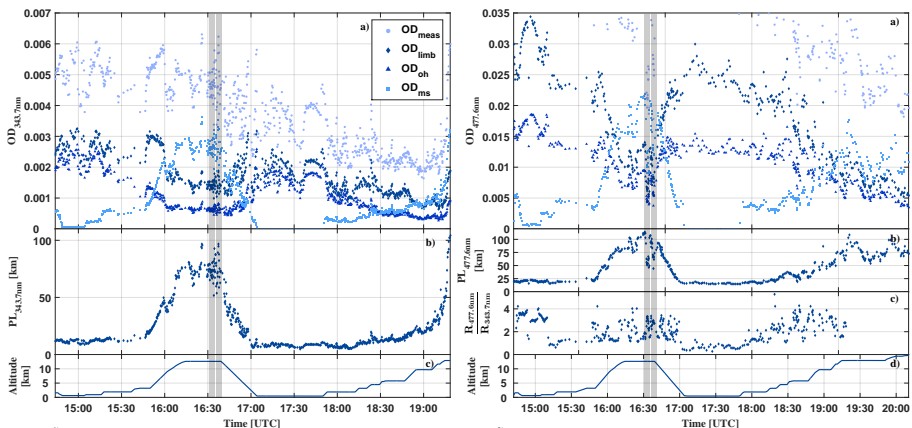

**Figure 2.** Attribution of the total measured optical depth of $O_4$ at 343.7 nm (panel (1) on the left) and at 477.6 nm (panel (2) on the right) to the various fractions, i.e., absorption from overhead the aircraft ($OD_{oh}$), from multiple scattering below the aircraft ($OD_{ms}$), within the line-of-sight of the telescopes ($OD_{limb}$), and the total measured optical depth of $O_4$ ($OD_{meas}$) for the HALO flight on Sept. 16, 2014 (AC11) (panels (1a), (2a)). Average line-of-sight photon path length inferred from the measured $OD_{meas}$ at 343.7 nm and 477.6 nm, respectively, as a function of the flight time (panels (1b), (2b)). The colour ratio $\frac{R_{343.7}}{R_{477.6}}$ of the measured radiances at 343.7 nm versus 477.6 nm is shown in panel (2c). The flight trajectory of the HALO aircraft is shown in panels (1c) and (2d). The sharp drop of the optical depth between 16:32–16:34 UTC and 16:35–16:39 UTC (grey bars) coincides with the passage of a cumulonimbus cloud in front of the telescopes. For details see the text.

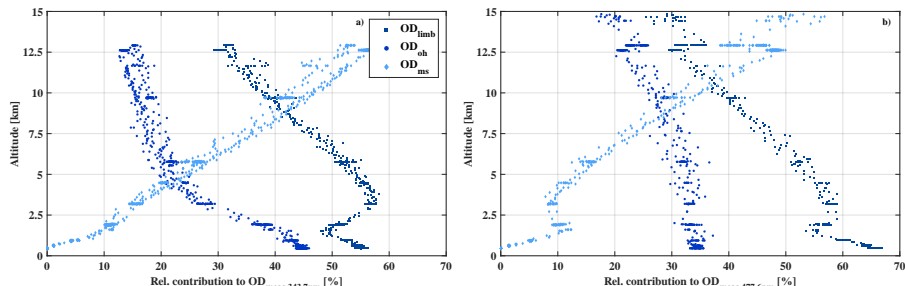

**Figure 3.** Relative contribution to the limb measured optical depth of $O_4$ at 343.7 nm (panel (a)) and at 477.6 nm (panel (b)) due to the absorption overhead the aircraft ($OD_{oh}$), from multiple scattering ($OD_{ms}$), and within the line of sight of the telescopes ($OD_{limb}$) for the HALO flight on Sept. 11, 2014 (AC11). For details see the text.

seen at the top of the planetary boundary layer at approximately 2 km, where often stratocumulus clouds prevail over the Amazon. Further, a maximum of $OD_{limb}$ is visible at about 4 km, most probably a result of the increasing horizontal visibility with altitude and the moderate contribution of reflected multiple scattered photons from the lower atmosphere to the total $O_4$ absorption. In the upper troposphere, the relative contribution of $OD_{limb}$ to $OD_{meas}$ decreases while the effective horizontal

light paths become longer and reach maximum values of over 75 km. But given all uncertainties in the details of the actual





atmospheric aerosol content, the cloud structure and coverage and their optical properties, it is more reasonable to speak here of an indication rather than a determination of the photon path lengths.

Fortunately, these uncertainties do not propagate into the uncertainty of the $\alpha$-factor ratio and hence into the determination of $[X]_j$, provided the gases X and P have similar profile shapes (Knecht, 2015; Hüneke et al., 2017; Stutz et al., 2017). From the above discussion, it also becomes clear that our air-borne UV/vis limb measurements average over some atmospheric volume, which precludes direct comparisons with in situ measured quantities on spatial scales smaller than the current averaging volume. This is of relevance to our study, for example when adding information from in situ measurements like CO to the analysis of our data.

Summing-up all described uncertainties, the precision error of the combined methods is calculated according to eq. (1) as

$$[\Delta X]_j = \sqrt{\left(\frac{\Delta\alpha_{R,j}}{\alpha_{R,j}}\right)^2 + \left(\frac{\Delta S_{X,j}}{S_{X,j}}\right)^2 + \left(\frac{\Delta S_{P,j}}{S_{P,j}}\right)^2 + \left(\frac{\Delta P_j}{P_j}\right)^2} \cdot [X]_j, \tag{2}$$

where $\Delta\alpha_{R,j}$ accounts for (a) random Mie extinction, (b) small scale variability of the target and scaling gas mixing ratios at flight level and (c) uncertainty in the sampling contribution as discussed in fig. 2 and 3. The slant column errors $\Delta S_{X,j}$ and $\Delta S_{P,j}$ account for (a) uncertainty of the inferred fraunhofer reference SCD and (b) the DOAS Fit error. $\Delta P_j$ accounts for the uncertainty of the in situ mixing ratio of the scaling gas, which in the case of $O_4$ scales with the air density and is accordingly limited by the measurement uncertainty of pressure and temperature. Additionally, the systematic errors of the individual absorption cross sections need to be added to $\Delta S_{X,j}$ according to Table 1. An extensive discussion of the total error budget of the $O_4$ scaling can be found in Hüneke (2016). While $\Delta P_j$ and $\Delta\alpha_{R,j}$ contribute approximately constantly to the total error budget, $\Delta S_{X,j}$ strongly increases with decreasing slant column density. For mixing ratios below 1 ppb ($CH_2O$), 0.15 ppb ($C_2H_2O_2$), and 1.3 ppb ($C_2H_4O_2^*$), the total precision error is strongly dominated by $\Delta S_{X,j}$. For higher mixing ratios, $\Delta\alpha_{R,j}$ and $\Delta S_{X,j}$ contribute in equal parts in the cases of $C_2H_2O_2$ and $C_2H_4O_2^*$. For $CH_2O$, $\Delta\alpha_{R,j}$ is the major error factor for mixing ratios above 1 ppb, and $\Delta S_{X,j}$ only makes up 30 % of the total error. For all gases, $\Delta P_j$ never exceeds 7 % of the total precision error. Due to the strong dependence of $\Delta S_{X,j}$ on the slant column density and the exponentially decreasing vertical profiles of the gases, the resulting total precision errors $[\Delta X]_j$ are strongly altitude dependent (fig. 4). They range from 16 % to 100 % for $CH_2O$, 17 % to 100 % for $C_2H_2O_2$, and 16 % to 100 % for $C_2H_4O_2^*$. This study focuses on measurements below 2 km altitude, with mean total errors of the analysed biomass burning events on the order of 21 % for $CH_2O$ (upper and lower limit of the total precision errors are 350 ppt and 740 ppt), 25 % for $C_2H_2O_2$ (total errors from 30 ppt to 45 ppt), and 22 % for $C_2H_4O_2^*$ (total errors from 240 ppt to 560 ppt).

## 2.2 AMTEX CO Measurements

In situ CO measurements were performed using an Aero-Laser 5002 vacuum UV resonance fluorescence instrument (Gerbig et al., 1996, 1999) installed in the cabin of the HALO aircraft. Main components of the CO instrument include a fluorescence chamber, resonance lamp, optical filter, two dielectric mirrors, and a pump. Atmospheric air is sampled through a backward-facing inlet mounted on top of the HALO fuselage. CO molecules in the ambient air are excited in the chamber by radiation from a radio frequency discharge resonance lamp. The wavelength range (centre line 151 nm with a bandwidth of 9 nm) is





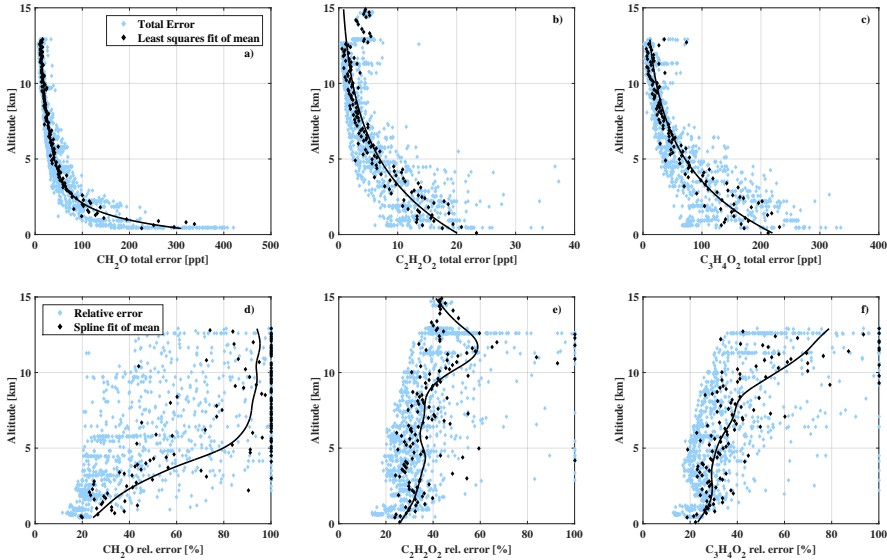

**Figure 4.** Relative and absolute total precision error of all measurements as a function of the measurement altitude.

selected by the optical filter and the mirrors. The fluorescence signal is detected by a photomultiplier. The flow rate of the
sample air through the fluorescence chamber is set to $40\ \mathrm{mL\,min^{-1}}$ (STP). The sample air is dried with a SICAPENT trap
(granulated phosphorus pentoxide drying agent) to eliminate a cross-sensitivity to water vapour.

The CO measurements are made at 1 Hz with a precision of 1.5 ppb. The accuracy is given by the precision and systematic
errors caused by the drift of the background signal and sensitivity of the instrument, and the uncertainty of the CO calibration
standard. The total accuracy amounts to $1.5\ \mathrm{ppb} \pm 2.4\%$ (Gerbig et al., 1999). The CO instrument has been deployed during
many aircraft campaigns using the DLR Falcon (e.g. Huntrieser et al. (2016a, b)) and was used for the first time used on board
HALO during the ACRIDICON-CHUVA campaign (Wendisch et al., 2016).

## 3 Measurements

The ACRIDICON-CHUVA aircraft campaign took place from the operational basis Manaus (Brazil) within the period Septem-
ber and early October 2014 with a total of 14 scientific flights over the Amazon region. An overview of all ACRIDICON-
CHUVA flight tracks can be found in Wendisch et al. (2016), fig. 6. The campaign aimed to improve the understanding of
tropical deep convective clouds and their interaction with biogenic and anthropogenic aerosols, and also included precipita-
tion formation and air pollution studies. A detailed description of the campaign objectives, the participating instruments and
research groups is given in Wendisch et al. (2016).

Our study requires simultaneous measurements of $CH_2O$, $C_2H_2O_2$, $C_3H_4O_2^*$ and $O_4$ (performed by the mini-DOAS spec-
trometers in the UV and visible spectral ranges as described in sect. 2.1) as well as CO measurements (performed by the





AMTEX instrument, see sect. 2.2), but either of the instruments occasionally failed. Therefore, only for a subset of flights joint data is available. Further, since a part of the flights focused on objectives other than near surface air pollution, here we only present data from four out of the 14 ACRIDICON-CHUVA flights, i.e. those of Sept. 11th (AC09), 16th (AC11), 18th (AC12), and 19th (AC13) in 2014. Nevertheless, the data collected during these flights provides valuable information on the different
measured pollutants, their sources, atmospheric transformation and transport in pristine background air in the troposphere as well as fresh and aged air affected by biomass burning in the Amazon.

Each of these measurement flight took seven to eight hours, typically occurring between 14 UTC and 22 UTC, i.e. during daytime. The long flight time supported the sampling of a large geographical area of the rain forest during the four flights, reaching from approximately 6° S to 8° N, and 49° W to 63° W. The detailed flight tracks and respective altitudes are shown
in fig. 5. During each of these flights, the probed altitudes ranged from several 100 m up to 15 km, with flight sections inside the planetary boundary layer overpassing large agricultural areas (AC12 and AC13) as well as the rain forest and the Amazon river delta (AC09 and AC11, fig. 5).

Due to the relatively large, but patchy cloud cover over the Amazon, the UV and vis limb radiances varied significantly during the flights. Therefore, typical integration times of the measured co-added sets of skylight spectra range from seconds
(under clear sky conditions) to several minutes (during cloudy sky and hazy conditions). In consequence, the along track resolution of the measurements was typically several kilometres in addition to the horizontal resolution perpendicular to the aircraft determined by radiative transfer (sect. 2.1). For the biomass burning events AC11-1 to AC11-3 (Table 3 and sect. 5.4), a two-dimensional visualization of the resulting averaging (or probed) areas per single measurement is indicated by the blue areas in fig. 6. The identification of the atmospheric measurement conditions and in particular of each biomass burning event is
based on visual imagery using the recorded video data stream (fig. 7). In the following, only those measurements are claimed to be affected by biomass burning, for which the actual plumes are visible in the recorded images and are in the telescope's line of sight. These events are listed in Table 3. Depending on the flight pattern and the integration time of the spectra, some of these plumes were intercepted more than once. In total, eight distinct plumes were probed during twelve individual measurements, all of which were recorded below 2 km flight altitude.

The MODIS instrument on board the Terra and Aqua satellites (MCD14, collection 6, available online from https://worldview.earthdata.nasa.gov/, doi:10.5067/FIRMS/MODIS/MCD14DL.NRT.006) reports additional fires in the vicinity of the aircraft trajectory, which were not directly located within in the field of view of the mini-DOAS telescopes, nor within the larger field of view of the camera. Therefore, these events are not more closely investigated in our analysis. However, the emissions of these fires are likely to
have created an atmospheric background generally affected by differently aged biomass burning emissions within the boundary layer (fig. 6). Further insights into the recent photochemical past of the air masses are provided by 3 h backward air mass trajectories calculated using the READY (Real-time Environmenral Applications and Display sYstem) website of the NOAA HYSPLIT model (https://www.ready.noaa.gov/HYSPLIT.php) (Stein et al., 2015; Rolph et al., 2017). Exemplary 3 h backward trajectories are shown for events AC11-1.2 to AC11-3.2 in fig. 6. They indicate, that prior to detection all air masses resided
well within the boundary layer, but did not directly pass over any additional reported fire.



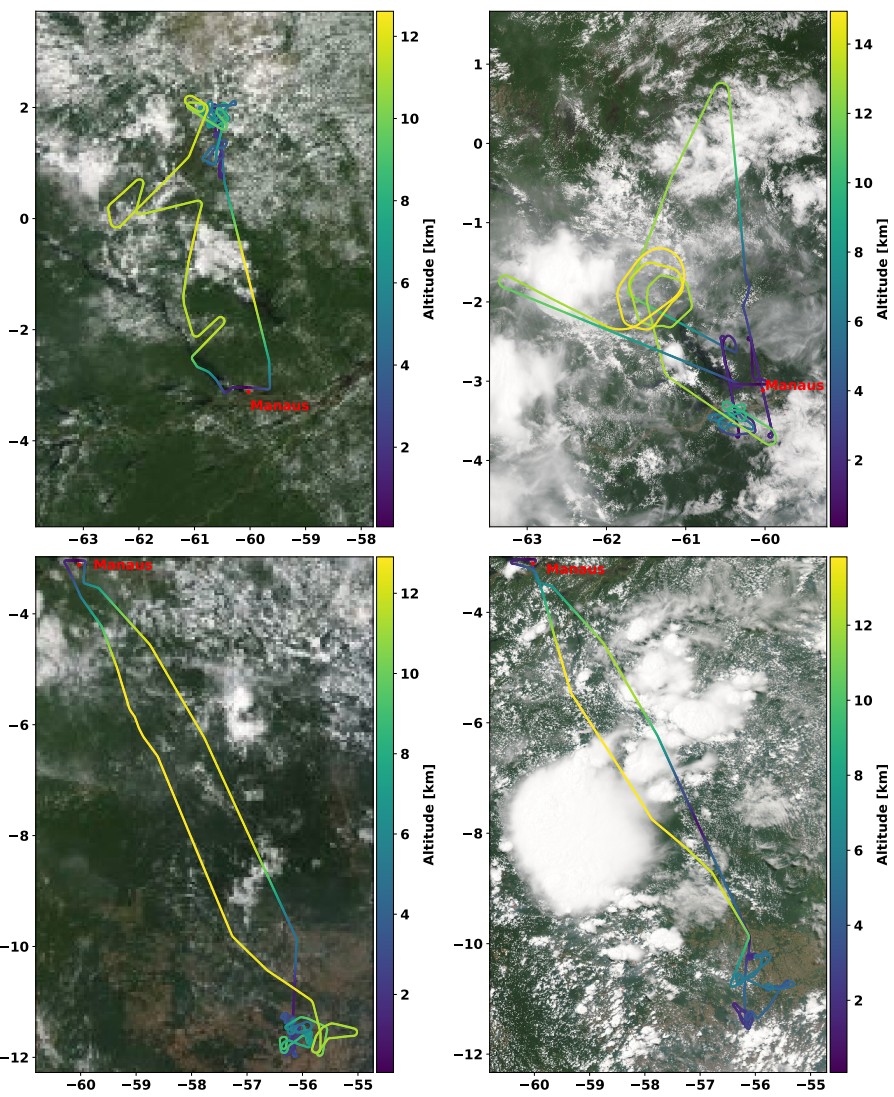

**Figure 5.** Tracks of the ACRIDICON-CHUVA flights (a) on Sept. 11, 2014, (b) on Sept. 16, 2014 (AC11), (c) on Sept. 18, 2014 (AC12), and (d) on Sept. 19, 2014 (AC13). All flights started and ended at the operational base Manaus airport in Brazil (red dot). The colour coding indicates the flight altitude. The MODIS satellite images are taken from NASA WORLDVIEW and are updated daily, see https://worldview.earthdata.nasa.gov/.

However, all identified biomass burning events show large and correlated enhancements of $CH_2O$, $C_2H_2O_2$, and $C_3H_4O_2^*$, with mixing ratios up to 4.0 ppb, 0.26 ppb, and 2.8 ppb, respectively (fig. 8). Due to the differently sized spatial resolutions of both instruments (sect. 2), the detected peaks in CO and those of $CH_2O$, $C_2H_2O_2$, and $C_3H_4O_2^*$ do not (and are not expected to) strictly correlate. Therefore, we refrain from directly estimating emission ratios with respect to CO.

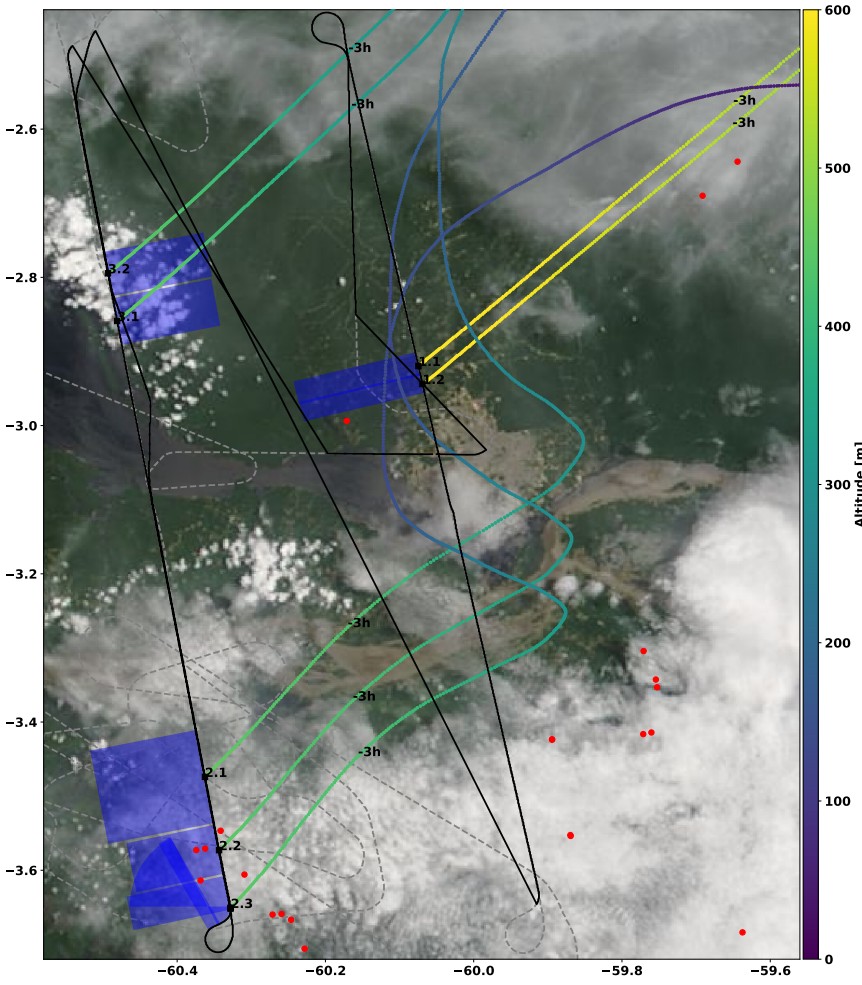

**Figure 6.** Zoom of the flight track for AC11 on Sept. 16, 2014. The flight tracks below/above 1000 m altitude are labelled by drawn/dashed lines, respectively. The take-off and landing of the aircraft was from Manaus airport (3.1° S, 60.0° W) located in the centre of the image. The biomass burning events detected by the MCD14 MODIS instrument on the Terra and Aqua satellites are indicated with red dots (not to scale). The biomass events probed by the mini-DOAS instrument are marked by black numbers next to the flight track, according to the labelling of Table 3. The blue rectangles approximately indicate the probed air masses in the horizontal plane, where the distance perpendicular to the flight track is given by the $O_4$ estimated photon path lengths (14 to 19 km), and the along track distance (3 to 15 km) by the spectrum integration time (33 to 166 s) multiplied by the aircraft ground speed (approximately 95 m s$^{-1}$) (sect. 2.1). For each biomass burning event, 3 h backward air mass trajectories were calculated and plotted colour-coded by the altitude of the airmass. The MODIS satellite image was taken from NASA WORLDVIEW, see https://worldview.earthdata.nasa.gov/.

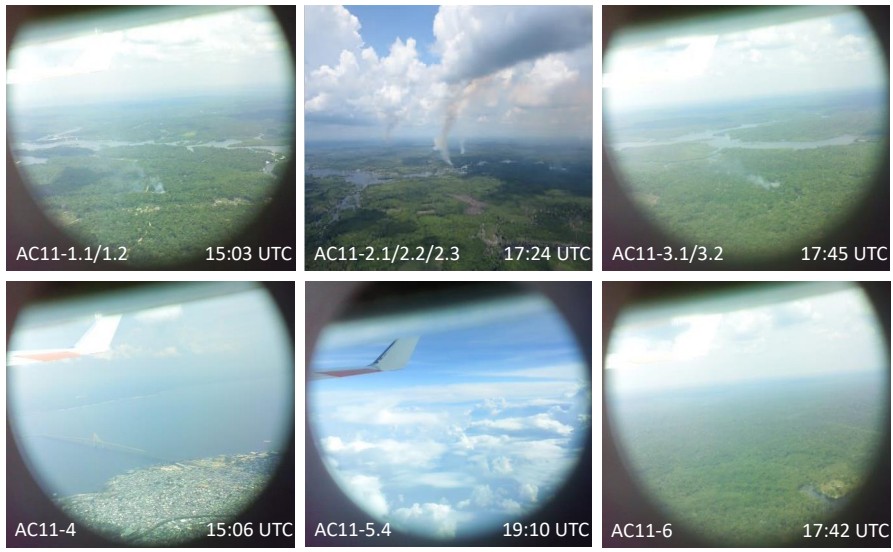

**Figure 7.** Images taken by the mini-DOAS camera during events 1–6 (as indicated in fig. 6 and 8) for the flight AC11 with the UTC time given in each image. The second image was taken from the respective flight report. The video camera points into the same direction as the three limb telescopes of the mini-DOAS spectrometer, however with a much larger field of view of $\sim 46°$ width.

## 4 Results

Figure 8 shows the inferred volume mixing ratios of $CH_2O$ (panel (a)), $C_2H_2O_2$ (panel (b)), $C_3H_4O_2^*$ (panel (c)), and CO (panel (d)) for flight AC11 on Sept. 16, 2014. The measured trace gas mixing ratios are the largest for flight sections within the planetary boundary layer (14:45 to 15:15 UTC and 17:05 to 17:45 UTC), and the lowest in the upper troposphere (16:15 to 16:45 UTC and 19:15 to 20:20 UTC). Notable are the peaks in CO while flying through the Manaus city plume (grey bars) or biomass burning plumes (red bars). Elevated mixing ratios of $CH_2O$, $C_2H_2O_2$, and $C_3H_4O_2^*$ can be related to biomass burning as well as to air masses of presumably aged background air (labelled with the green bars) as evidenced by the only moderately elevated CO. Here, the numbers (1) to (6) in panel (a) of fig. 8 denote biomass burning affected air (1–3), air of the Manaus city plume (4), pristine air of the upper troposphere (5), and boundary layer air above the tropical forest (6). Sky images of all six different situations are shown in fig. 7.

### 4.1 Vertical profiles

Below 450 m altitude, CO mixing ratios range from 88.6 ppb to 291.8 ppb (fig. 9, panel (a)), with a mean of [CO]=188.1 ppb. Within the planetary boundary layer, CO mixing ratios increase from 88.4 ppb to 2308 ppb (mean [CO] = 212.8 ppb), with peak mixing ratios above 500 ppb appearing when biomass burning plumes are passed directly. Lowest [CO] mixing ratios



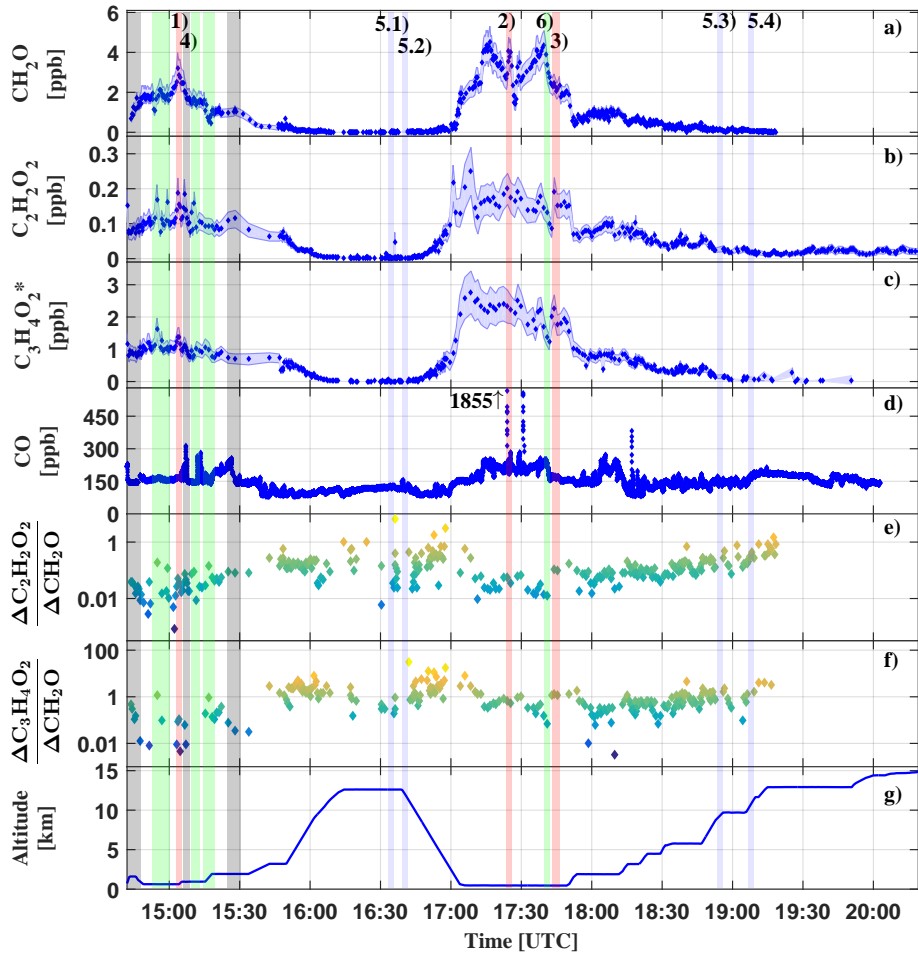

**Figure 8.** Measured mixing ratios of $CH_2O$ (panel (a)), $C_2H_2O_2$ (panel (b)), $C_3H_4O_2^*$ (panel (c)), CO (panel (d)), inferred $R_{GF} = \Delta C_2H_2O_2/\Delta CH_2O$ (panel (e)), $R_{MF} = \Delta C_3H_4O_2^*/\Delta CH_2O$ (panel (f)), and the height versus time trajectory for the HALO flight AC11 (panel (h)). All measurements within the Manaus city plume are marked by grey bars (e.g. event AC11-4). Examples for measurements in aged air masses of the upper troposphere are marked in blue (e.g. events AC11-5.1–5.4), and green for the general background atmosphere i.e. tropical forest affected air (e.g. event AC11-6). All measurements of biomass burning plumes are labelled in red (events AC11-1, AC11-2 and AC11-3). The event numbers correspond to the labelling in Table 3 and fig. 7. The colour coding in panels (e)), and (g) corresponds to the inferred $R_{GF}$ or $R_{MF}$ of each measurement, in order to emphasize the differences in the gas ratios otherwise not clearly distinguishable due to the logarithmic scale of the y-axis.

in the range of 55.1 ppb to 231.9 ppb (mean [CO] = 86.3 ppb) are found in the middle troposphere between 6 and 8 km
altitude. In the upper troposphere (12–14 km), CO increases from 52.5 ppb to 212.6 ppb (mean [CO] = 134.5 ppb). In the convective tropics, C-shaped CO profiles are a well-known phenomenon, caused by the rapid transport of near surface CO rich air by meso-scale convective systems into the upper troposphere (e.g., Brocchi et al. (2018); Krysztofiak et al. (2018)). When

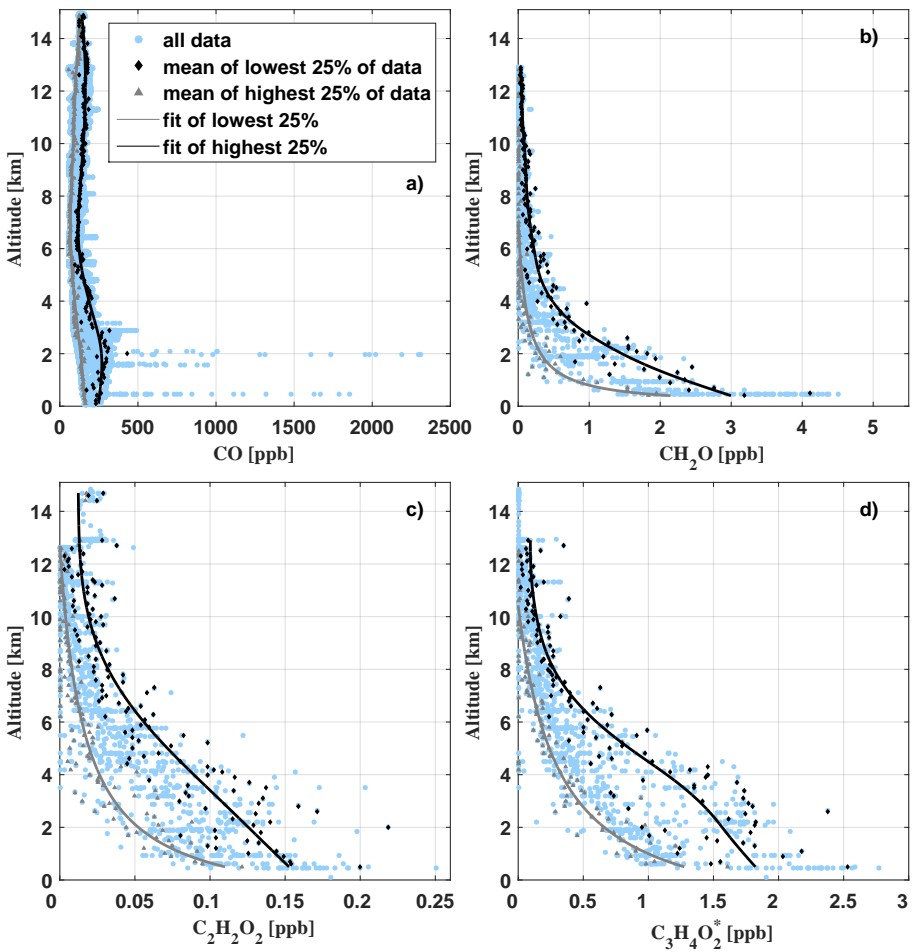

**Figure 9.** Vertical profiles of CO (panel (a)), $CH_2O$ (panel (b)), $C_2H_2O_2$ (panel (c)), and $C_3H_4O_2^*$ (panel (d)) as measured during flights AC09, AC11, AC12, and AC13 (blue circles). For AC09, there are no CO data available. The grey lines indicate a spline fit (panel (a)) and least squares fits (panel (b) to (d)) through the lower quartile of the data (grey triangles) in 100 m altitude bins. The black line shows corresponding fits through the mean of the upper quartile (black diamonds).

binning the data in 100 m altitude intervals and averaging over the lower and upper data quartiles within each interval, another maximum forms around the top of the boundary layer. Most likely, this suggests that over the Amazon biomass burning plumes often detrain their pollutants near the top of the boundary layer (cf. see for example the image at 17:24 UTC, fig. 7).

Inferred $[CH_2O]$ ranges from 4.5 ppb near the surface to mean mixing ratios of 30 ppt in the upper troposphere (above 10 km). Mixing ratios within the boundary layer and middle troposphere vary significantly (fig. 9, panel (b)). In part, this might be a manifestation of the direct emission of $CH_2O$ into the lower part of the atmosphere, or the in situ formation of $CH_2O$ during the oxidation of biogenic VOCs (BVOCs) as well as partly oxidized VOCs (OVOCs). Figure 10 puts our $CH_2O$ measurements over the Amazon into the context of previously published $CH_2O$ observations all over the world. Within this





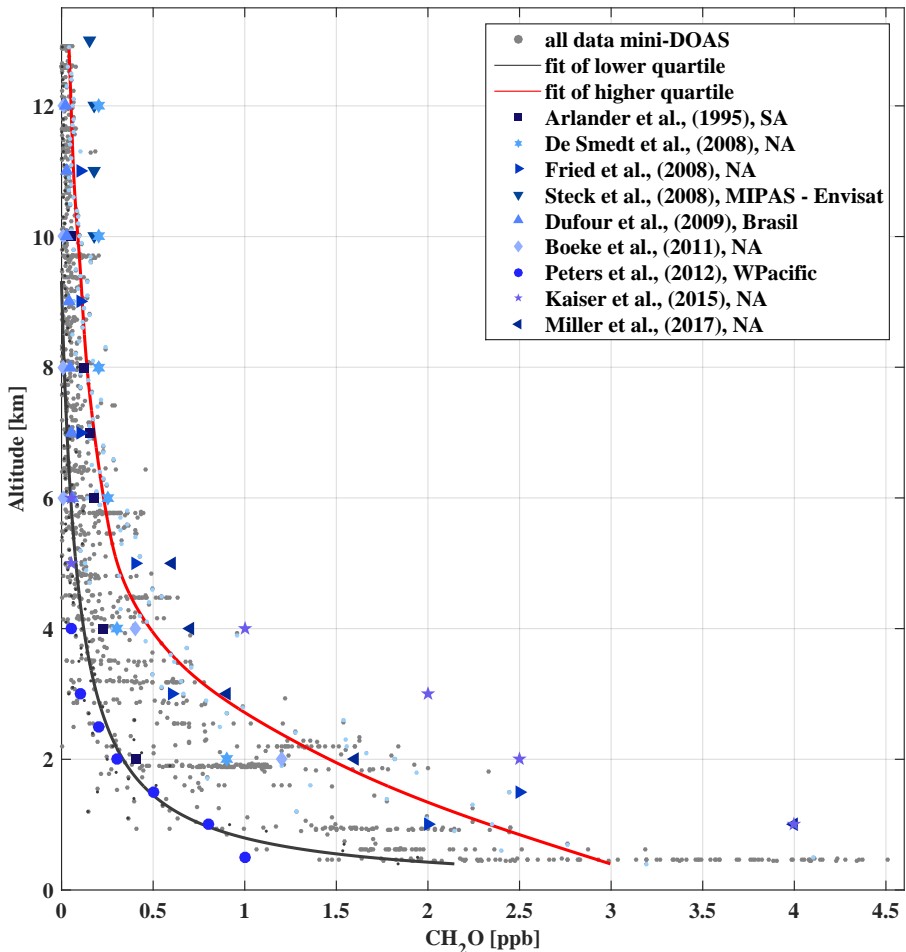

**Figure 10.** Comparison of different $CH_2O$ measurements available from the literature (Arlander et al., 1995; De Smedt et al., 2008a; Fried et al., 2008; Dufour et al., 2009; Steck et al., 2008; Boeke et al., 2011; Peters et al., 2012; Kaiser et al., 2015; Chan Miller et al., 2017) with those of the present study (black). The geographical regions of the measurements are Brazil, South America (SA), North America (NA), West Pacific (WPacific) and MIPAS-Envisat measurements covering orbit 8164, 14° S/46° W.

comparison, it is remarkable that the lowest $CH_2O$ mixing ratios measured over the Amazon approximately agree with $CH_2O$ measured within the boundary layer and middle troposphere over the West Pacific by Peters et al. (2012). Our measurements in the upper troposphere compare well to findings from Dufour et al. (2009) and Boeke et al. (2011) over Brazil and North America, respectively. The lowest mixing ratios measured in the middle and upper troposphere may reflect $CH_2O$ formed during the oxidation of mostly $CH_4$ and, if available, some residual VOCs, OVOCs, and BVOCs (cf., see fig. 9, panel (c) for $C_2H_2O_2$ and panel (d) for $C_3H_4O_2^*$), since their heterogeneous removal is certainly slower than the $CH_2O$ photochemical lifetime of only a few hours (Frost et al., 2002). In contrast, the largest $CH_2O$ mixing ratios measured over the Amazon at any altitude are appear slightly smaller than the average, and significantly smaller than the maximum mixing ratios reported from





the joint SENEX aircraft observations of $CH_2O$ and $C_2H_2O_2$ over North America (Kaiser et al. (2015); Chan Miller et al.
315 (2017)).

$C_2H_2O_2$ mixing ratios range from 85 ppt to 250 ppt below 500 m altitude and decrease in the upper troposphere (12–
14 km) to mixing ratios from 5 ppt (our detection limit) to about 49 ppt. Close to the ground, our results are comparable
to $C_2H_2O_2$ mixing ratios found over North America by Kaiser et al. (2015) and Chan Miller et al. (2017), but much larger
than those measured in pristine air masses of the South Pacific (7–23 ppt) by Lawson et al. (2015). Evidently, besides the
$C_2H_2O_2$ formation during the oxidation of BVOCs (Wennberg et al., 2018), a fraction of the near surface and boundary
layer $C_2H_2O_2$ is certainly due to direct emission by biomass burning (Zarzana et al., 2018; Andreae, 2019). In the middle
troposphere, our inferred mixing ratios of $C_2H_2O_2$ are significantly larger than observations during SENEX (which reach to
approximately 6 km altitude) and are also larger in the upper troposphere above 6 km when comparing to data extrapolated
from SENEX. Between 6 and 14 km altitude, $C_2H_2O_2$ enhancements of up to $\approx 63$ ppt become apparent when compared to
either the SENEX data or the binned lower quartile of our data. Such elevated mixing ratios of $C_2H_2O_2$ in the middle and
upper troposphere point to an efficient vertical transport of $C_2H_2O_2$ and its precursors VOCs, OVOCs, and BVOCs from their
emission sources at the ground (Andreae et al., 2001). Further, $C_2H_2O_2$ (and $C_3H_4O_2^*$) present at higher atmospheric altitudes
may serve as a marker for the formation of ISOPOO (isoprene peroxy radicals), ISOPOOH (oligomer hydroxyhydroperoxides)
and finally IEPOX (isoprene epoxydiols) within the isoprene oxidation chain (Wennberg et al., 2018). In fact, IEPOX mediated
SOA formation in the upper troposphere over the Amazon has been reported from observations made within the framework of
the ACRIDICON-CHUVA project (Andreae et al., 2018; Schulz et al., 2018).

The detection of $C_3H_4O_2^*$, here taken as the sum of methylglyoxal and larger carbonyls as described previously, appears elu-
sive due to the spectral interference among the different species. Weighting of the measured total absorption for $C_3H_4O_2^*$ with
the relative absorption cross sections of the inferred $C_3H_4O_2$, $C_4H_6O_2$ (biacetyl), and $C_4C_8O_4$ (acetylpropionyl) (Zarzana
et al. (2017), fig. S4) may be indicative of the relative abundance of each species. As in Zarzana et al. (2017), we recommend
a weighting factor of $2.0 \pm 0.5$ by which the inferred $C_3H_4O_2^*$ needs to be divided for indicative $C_3H_4O_2$ mixing ratios.

Due to the lack of previous atmospheric $C_3H_4O_2^*$ (or $C_3H_4O_2$) measurements, it is more difficult to validate our inferred
$C_3H_4O_2^*$ profile (fig. 9, panel (d)). As for the other measured hydrocarbons, the largest $C_3H_4O_2^*$ mixing ratios between 1.2 ppb
and 2.8 ppb (mean $[C_3H_4O_2^*]$=1.6 ppb) are found near the ground and within the planetary boundary layer. Below 1 km
altitude, our results of 0.8 ppb to 2.8 ppb (mean $[C_3H_4O_2^*]$=1.5 ppb) are considerably larger than recently reported $C_3H_4O_2$
mixing ratios (28 ppt to 365 ppt) from a Mediterranean site with intense biogenic emissions and low levels of anthropogenic
trace gases (Michoud et al., 2018), as well as measurements at Cape Grim ($28 \pm 11$ ppt) and the Chatham Rise ($10 \pm 10$ ppt)
in pristine marine air (Lawson et al., 2015). Therefore, we expect a major fraction of the enhanced $C_3H_4O_2^*$ to be related to
the oxidation of BVOCs and biomass burning (e.g., Andreae and Merlet (2001); Akagi et al. (2011); Stockwell et al. (2015);
Zarzana et al. (2017, 2018); Andreae (2019)). Evidence that the former is overall the more relevant process (as compared to
direct emissions by biomass burning) in the Amazonian troposphere is also provided by the relatively compact clustering of
the inferred $C_3H_4O_2^*$ along the fit of the lower quartile of the data.





## 5 Discussion

### 5.1 Vertical profiles and precursor VOCs

Above the boundary layer, significant enhancements in $C_2H_2O_2$ and $C_3H_4O_2^*$ mixing ratios above the inferred backgrounds are observed (fig. 9, panels (c) and (d)). After applying a correction factor of $2.0 \pm 0.5$ to the measured $C_3H_4O_2^*$ mixing ratios as discussed above, methylglyoxal exceeds the inferred glyoxal mixing ratios by up to a factor of five for all measurements. Because of many long flight sections above the remote rain forest, which were to our knowledge at least partly free of fires and plumes, as well as the comparatively different shape of the formaldehyde profile, not all of these enhancements can be

attributed to direct emissions from biomass burning.

As their dominant source, isoprene globally accounts for 67 %, 47 %, and 79 % of the yearly emissions of formaldehyde, glyoxal, and methylglyoxal (Fu et al., 2008), respectively, leading to their generally correlated vertical profiles. As argued by Fu et al. (2008) and Fu et al. (2019), the oxidation of isoprene under low $NO_x$ conditions may be delayed by hours or even several days. This prolonged lifetime combined with the efficient vertical atmospheric transport in the tropics leads to

isoprene oxidation above the boundary layer. As a consequence, we suspect the in situ formation of glyoxal and methylglyoxal and finally formaldehyde from isoprene oxidation to lead to the observed enhanced mixing ratios in the free troposphere. The methylglyoxal enhancements in the middle troposphere are approximately five times larger than the observed glyoxal mixing ratios. Molar yields of glyoxal and methylglyoxal from second and third generation isoprene oxidation combined, are 6.2 % and 34 %, respectively (Fu et al., 2008). Therefore, we expect the in situ formation of methylglyoxal from isoprene oxidation

to be approximately five times larger than the respective glyoxal production. In accordance with our measurements, this should lead to five times larger methylglyoxal mixing ratios compared to glyoxal in the free troposphere.

Despite their common dominant precursor, the secondary sources of the gases are different (PAN for formaldehyde, acetylene for glyoxal, and acetone for methylglyoxal (Fu et al., 2008)), and might therefore differently contribute to our measurements. While acetylene is mostly an anthropogenic emitted trace, acetone additionally has direct biogenic sources. Both gases have a

long lifetime of up to 18 and 22 days, respectively (Fu et al., 2008). Therefore, vertically transported acetylene from ground-based combustion processes might be a further source of the observed glyoxal production in the free troposphere. While acetone is also emitted in biomass burning processes (one fifth of the total acetone budget (Pöschl et al., 2001)), an equally large source of acetone emissions is decomposing plant material ($10^{-4}\,\mathrm{g\,g^{-1}}$ (Warneke et al., 1999)). The remaining acetone is to a large part an oxidation product of hydrocarbons and also an oxidation product of isoprene under low $NO_x$ conditions, which are

typical for rain forests (Warneke et al., 2001). Mean acetone mixing ratios in the boundary layer above tropical rain forests are on the order of $2\,\mathrm{nmol\,mol^{-1}}$ (Pöschl et al., 2001). Besides isoprene oxidation, a fraction of the observed methylglyoxal enhancements in the free troposphere might therefore be a direct or secondary (through the oxidation of hydroacetone) product of acetone oxidation above the boundary layer.

It is remarkable, that even though formaldehyde, glyoxal, and methylgloxal have a common dominant source (isoprene)

and sink (photolysis), the relative profile shapes are different. Especially above the boundary layer, the inferred formaldehyde mixing ratios decrease faster with increasing altitude than the vertical profiles of glyoxal and methylglyoxal do (fig. 9, panel





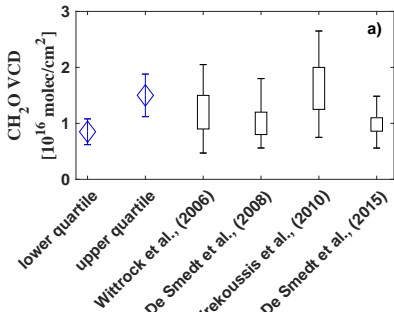 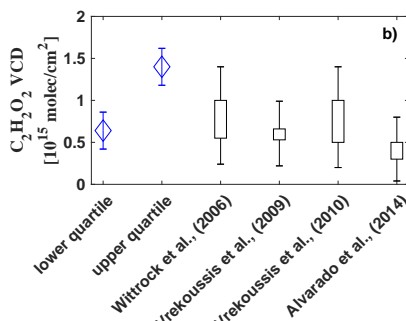

**Figure 11.** Comparison of the integrated total column amounts of $CH_2O$ and $C_2H_2O_2$. The vertical bars indicate the range of the observed vertical column amounts as reported in the respective studies. The uncertainty range for Wittrock et al. (2006) and Alvarado et al. (2014) was estimated based on Vrekoussis et al. (2009).

(b)). It is thus to question whether the relative contribution to the total sink of photolysis and reactions with OH radicals in a low-$NO_x$, high-VOC, and low-$HO_x$ environment or the different efficiencies regarding the uptake into aerosols or cloud particles may alter the relative profile shapes of the three species. The water solubility of the gases is largely different ($H^{cp} = 32$

mol m$^{-3}$ Pa$^{-1}$ for formaldehyde, $H^{cp} = 4100$ mol m$^{-3}$ Pa$^{-1}$ for glyoxal, and $H^{cp} = 370$ mol m$^{-3}$ Pa$^{-1}$ for methylglyoxal, see Sander (2015)) and significantly smaller for formaldehyde than glyoxal. Heterogeneous uptake may therefore not provide an explanation for the relative depletion of formaldehyde over gyloxal and methylglyoxal in the middle troposphere.

### 5.2  Comparison with satellite measurements

As neither $CH_2O$ nor $C_2H_2O_2$ have relevant stratospheric sources, the integration of the profiles yields their total vertical

column density (VCD). The uncertainty of the integrated profiles follows from the altitude weighted total error of our measurements as displayed in fig. 4. We compare our findings to measurements from GOME (Global Ozone Monitoring Experiment), SCIAMACHY (SCanning Imaging Absorption spectroMeter for Atmospheric CHartographY), OMI (Ozone Monitoring Instrument), and GOME-2, which combined provide global $CH_2O$ and $C_2H_2O_2$ observations covering more than a decade (fig. 11).

Integration of the $CH_2O$ profiles yields total column amounts of $(0.9 \pm 0.2) \cdot 10^{16}$ molec cm$^{-2}$ (lower quartile as plotted

in fig. 9) to $(1.5 \pm 0.4) \cdot 10^{16}$ molec cm$^{-2}$ (upper quartile), with a mean of $(1.1 \pm 0.3) \cdot 10^{16}$ molec cm$^{-2}$ (all data, fig. 11, a). Satellite observations generally report enhanced $CH_2O$ VCDs over regions with large biogenic emissions, especially over the Amazon Basin, with maximal enhancements during the dry season. Monthly means of $CH_2O$ VCDs measured by GOME and SCIAMACHY in the years 1996–2007 over the Amazon report maximal enhancements of $1$–$2 \cdot 10^{16}$ molec cm$^{-2}$ during

the dry season and much smaller vertical column densities of $0.8 \cdot 10^{16}$ molec cm$^{-2}$ during the wet season (De Smedt et al., 2008b). Our inferred $CH_2O$ vertical column densities based on the lower quartile agree well with their measurements during the wet season, while our integrated profile of the upper quartile lies within the observations during the dry season. Our results agree equally well with SCIAMACHY observations in 2005 over South America with an annual mean $CH_2O$ VCD





of approximately $1.2 \cdot 10^{16}$ molec cm$^{-2}$ as reported by Wittrock et al. (2006) and also agree with GOME-2 observations, for

which Vrekoussis et al. (2010) report CH$_2$O VCDs of 1.2 to $1.6 \cdot 10^{16}$ molec cm$^{-2}$ over the Amazon Basin for the years 2007 to 2008. Finally, yearly averaged CH$_2$O from OMI and GOME-2 observations over Brazil between 2007 and 2013 by De Smedt et al. (2015) agree with our mean of $1.1 \cdot 10^{16}$ molec cm$^{-2}$.

The integrated C$_2$H$_2$O$_2$ profiles range from $(0.6 \pm 0.2) \cdot 10^{15}$ molec cm$^{-2}$ (lower quartile) to $(1.4 \pm 0.4) \cdot 10^{15}$ molec cm$^{-2}$ (upper quartile), with a mean of $(1.0 \pm 0.3) \cdot 10^{15}$ molec cm$^{-2}$. Based on SCIAMACHY measurements in the years 2002 to

2007, Vrekoussis et al. (2009) report seasonal mean C$_2$H$_2$O$_2$ VCDs of $0.5 \cdot 10^{15}$ molec cm$^{-2}$ over northern South America in autumn (fig. 11, b). In good agreement with our lower limit, Wittrock et al. (2006) report slightly higher SCIAMACHY C$_2$H$_2$O$_2$ observations of approximately $0.6$–$0.7 \cdot 10^{15}$ molec cm$^{-2}$ over North Brazil for the year 2005. Corresponding to our mean VCD, Vrekoussis et al. (2010) further report average VCDs of $0.5 \cdot 10^{15}$ molec cm$^{-2}$ with maximal enhancements of $1 \cdot 10^{15}$ molec cm$^{-2}$ based on GOME-2 measurements over South America in the years 2007 to 2008. Observations from OMI

yield slightly lower monthly mean C$_2$H$_2$O$_2$ VCDs of $0.3 \cdot 10^{15}$ molec cm$^{-2}$ (Alvarado et al., 2014). The VCD based on the upper quartile, which is indicative of the large amount of direct glyoxal emissions from various biomass burnings into the boundary layer, is larger than, but still within the uncertainty range of the SCIAMACHY and GOME-2 observations.

### 5.3 $R^*_{XF}$ above the Amazon rain forest

As outlined above, the different air masses probed with the mini-DOAS and AMTEX instrument prevent a definition of emis-

sion ratios relative to CO (sect. 2.1 and 2.2). Instead, we infer the emission ratios $R^*_{GF}$ and $R^*_{MF}$ relative to CH$_2$O, as often done in remote sensing studies (Fu et al. (2008); Kaiser et al. (2015); Stavrakou et al. (2016); Zarzana et al. (2017, 2018), and others). According to the literature, the background uncorrected emission ratio is defined as

$$R^*_{XF} = \frac{[X]}{[CH_2O]}. \tag{3}$$

In this study, [X] is either [C$_2$H$_2$O$_2$] or [C$_3$H$_4$O$_2^*$], as often used in satellite and modelling studies related to hydrocarbon

precursors of the studied species (Fu et al. (2008); Kaiser et al. (2015), and others). $R^*_{XF}$ is calculated for each measurement and analysed with respect to the measurement altitude (fig. 12, panel (a) and (b)).

### 5.3.1 $R^*_{GF}$

$R^*_{GF}$ mostly remains smaller than 1.0, with a total mean of $\overline{R}^*_{GF}=0.35$, but reaches maximum values $> 5.0$ during several measurements. Small ratios $< 0.05$ and large ratios $> 1.0$ can be observed throughout all analysed flights and nearly all altitudes.

The comparison with Kaiser et al. (2015) shows, that the inferred $R^*_{GF}$ is notably larger than during most of their measurements over the south-eastern US. Our results are in much better agreement with the $R^*_{GF}$ MacDonald et al. (2012) inferred during measurements over an Asian rain forest. Their $R^*_{GF}$ in the range 0.2 to 0.7 agrees with most of our measurements as well as with $\overline{R}^*_{GF}$ of approximately 0.35. From measurements above the Kisatchie National Forest and the Mark Twain National Forest in the south-eastern US, Kaiser et al. (2015) conclude characteristically low $R^*_{GF}$ in pristine regions with strong isoprene

emissions, while regimes dominated by monoterpene emissions appear to have higher $R^*_{GF}$. Most of our measurements took



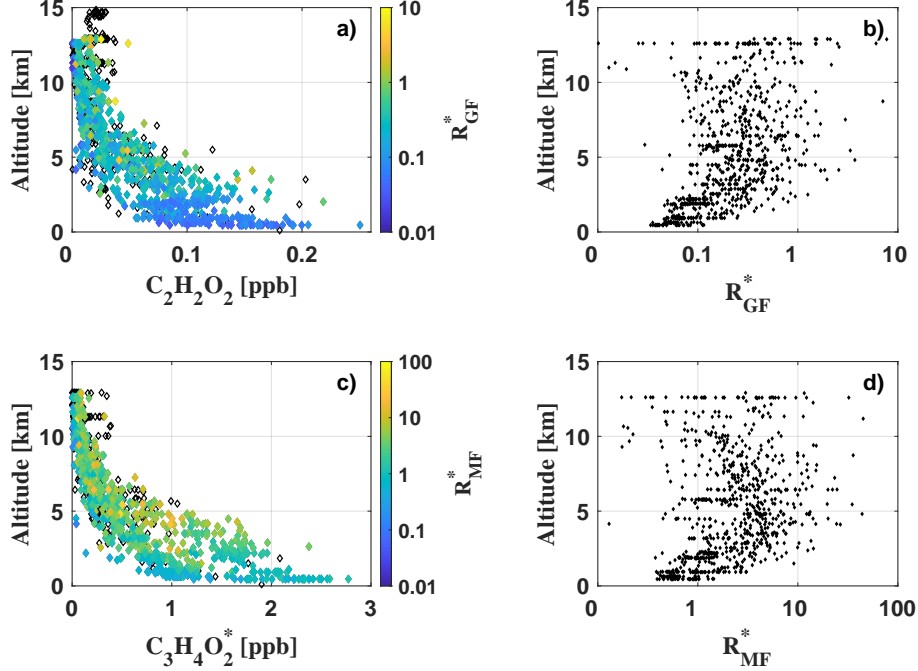

**Figure 12.** Vertical profiles of $C_2H_2O_2$ (panel (a)) and $C_3H_4O_2^*$ (panel (c)), colour-coded by the ratios $R_{GF}^*$ and $R_{MF}^*$, respectively, for each measurement during flights AC09, AC11, AC12, and AC13. Black diamonds indicate measurements with missing $CH_2O$ data (recorded with a different spectrometer) and consequently missing ratios. Panels (b) and (d) illustrate the vertical profiles of $R_{GF}^*$ and $R_{MF}^*$ throughout the troposphere. For panels (b) and (d) a logarithmic x-axis has been chosen for a better illustration of the altitude dependence of small $R_{GF}^*$ and $R_{MF}^*$, respectively.

place in pristine air masses above the rain forest and far away from any major population centre or other anthropogenic emission sources. Monoterpene and isoprene emissions over the central Amazon were studied in multiple investigations (e.g. Helmig et al. (1998) and Kesselmeier et al. (2000)). All these studies reported generally low mixing ratios of monoterpenes compared to isoprene, which composes 90 % of the total VOC budget in the Amazon (Kuhn et al., 2007). Emissions of monoterpenes,

like $\alpha$-pinene, on the other hand, are at least one order of magnitude smaller. Approximately the same relation was found for the total monoterpene emissions (Kuhn et al., 2007; Rizzo et al., 2010). Kaiser et al. (2015) report on approximately 1.2–2.0 times larger $C_2H_2O_2$ yield from $\alpha-$ and $\beta$-pinene oxidation with respect to $CH_2O$. Still, the significant predominance of isoprene emissions over the Amazon clearly compensate for this effect. We conclude that isoprene and not monoterpenes is the dominant precursor of $CH_2O$ and $C_2H_2O_2$ for our observations. Contrary to the measurements of Kaiser et al. (2015)

in isoprene rich regimes, the results indicate significantly elevated $R_{GF}^*$ in the troposphere over the Amazon. This is a direct consequence of the much lower $CH_2O$ mixing ratios as compared to their measurements over the south-eastern US. $CH_2O$ generally does not exceed 1–2 ppb (except for direct measurements in biomass burning or in the Manaus city plume), which is four times less than reported by Kaiser et al. (2015) for pristine air masses over the Mark Twain National Forest (ca. 8 ppb).





Our $C_2H_2O_2$ mixing ratios, on the other hand, are comparable to their findings and typically 100 ppt in the lower troposphere.
As a result, we obtain significantly higher $R^*_{GF}$ than Kaiser et al. (2015) for the pristine troposphere.

Based on the SENEX observations, Chan Miller et al. (2017) reported mean $R^*_{GF}$ of 0.024 below 1 km altitude. Interestingly, Chan Miller et al. (2017) obtained larger $R_{GF}$ of up to 0.06 when $NO_x$ was very low (0.1–0.5 ppb). Even though we are not able to infer $NO_x$ concentrations for all measurements due to instrument failures, we were able to measure $NO_2$ during all analysed flights. Inferred $NO_2$ mixing ratios from mini-DOAS measurements were very low during all flights. Mixing ratios
above 1 ppb were inferred only in rare cases and exclusively over direct emission sources like Manaus city and in biomass fire plumes. We conclude from these overall low $NO_2$ mixing ratios, that the measurements during the campaign were generally under very low $NO_x$ conditions. As described in detail by Chan Miller et al. (2017), this could lead to prompt glyoxal formation through isoprene peroxy radicals and thus slightly enhanced $R_{GF}$ compared to observations under high $NO_x$. Further, from our measurements over Manaus city, we cannot detect any sizeable influence of anthropogenic VOCs on the inferred $R^*_{GF}$. During
several low overpasses of the city (< 2 km flight altitude), $R^*_{GF}$ did not show any significant difference to measurements distant from anthropogenic emission sources. During the measurements in the city plume, the relative increase of $CH_2O$ and $C_2H_2O_2$ mixing ratios was similar, leading to consistent $R^*_{GF}$.

Finally, the vertical profile of $R^*_{GF}$ indicates slightly elevated ratios above the boundary layer and in the free troposphere (fig. 12, panel (b)). Within the boundary layer, $R^*_{GF}$ remains approximately constant. Both features were previously observed
by Kaiser et al. (2015) up to 6 km, however less pronounced. Above 10 km, our results indicate a slight decrease of $R^*_{GF}$. When discussing these observations, one has to keep in mind the very low $CH_2O$ and $C_2H_2O_2$ concentrations in the upper troposphere and the increasing influence of noise on the inferred mixing ratios. Above 6 km, we observe mean $CH_2O$ and $C_2H_2O_2$ of only $54 \pm 40$ ppt and $15 \pm 5$ ppt, respectively. Still, the increase in $R^*_{GF}$ between ~2–10 km appears most pronounced right above the boundary layer at 2 km, where $CH_2O$ and $C_2H_2O_2$ mixing ratios are still significantly above the
detection limits. Notably, the correlation of $C_2H_2O_2$ and $CH_2O$ is higher within the boundary layer. In the free troposphere, $R^*_{GF}$ varies on average by 60 % among the different measurements in the same altitude range. Following the argumentation of Kaiser et al. (2015), this behaviour can be attributed to the uniform mixing of $CH_2O$ and $C_2H_2O_2$ within the boundary layer.

### 5.3.2 $R^*_{MF}$

Generally, $R^*_{MF}$ appears approximately 10 times larger than $R^*_{GF}$ (fig. 12, panels (c) and (d)). $R^*_{MF} < 1$ are observed only
for a minority of the measurements and mostly for background $C_3H_4O_2^*$ concentrations. Larger $R^*_{MF} > 1$ are inferred at all altitudes, with a total mean $\overline{R}^*_{MF}=3.6$ and a maximum ratio $R^*_{MF,max}=47$. As discussed above for $R^*_{GF}$, the vertical profile of $R^*_{MF}$ indicates slightly larger $R^*_{MF}$ above the boundary layer and throughout the free troposphere. In the upper troposphere, $R^*_{MF}$ decreases again, leading to a concave curvature of the vertical profile for small $R^*_{MF}$, as previously described for $R^*_{GF}$. Several studies were published on the emission ratio of $C_3H_4O_2$ to $CH_2O$ (or CO) in biomass burning plumes (e.g. Hays
et al. (2002); Müller et al. (2016); Zarzana et al. (2017, 2018)). To our knowledge, comparable data for air masses of the free troposphere are still missing. As a result, we cannot thoroughly compare $R^*_{MF}$ to other findings, but focus our analysis on measurements within biomass burning plumes.





### 5.4 Normalised excess mixing ratios in biomass burning plumes and inferred emission factors

During the four measurement flights, a total of twelve biomass burning plume intercepts were identified based on video imagery
(e.g. fig. 7 for events AC11-1.1/1.2 and AC11-3.1/3.2). Most of the plume measurements occurred south of Manaus, and during
two extensive flight periods of AC11 and AC13 at altitudes below 2 km. The precise location of all plume intercepts during
flight AC11, and the related horizontal averaging kernels of the telescopes are given in fig. 6. As indicated by the backward
trajectories, three hours prior to detection all probed air masses moved well within the boundary layer, below 600 m altitude.
This time frame more than covers the atmospheric lifetimes of $CH_2O$, $C_2H_2O_2$ and $C_3H_4O_2$. Consequently, the measured
mixing ratios of these gases are a result of fresh emissions at the surface. Each biomass burning plume encounter correlates
with significant enhancements in $CH_2O$, $C_2H_2O_2$ and $C_3H_4O_2$ over the background by 13 to 400 %. Based on the backward
trajectories combined with video imagery and the short lifetimes of the gases, contributions from other major emission sources
(e.g. Manaus city) can largely be excluded. The exact timing, location and altitude of each plume intercept is given in Table
3. As indicated by the numbering in column 1, some of the plumes were probed more than once, giving a total of eight
different biomass burning plumes with 12 encounters (further on called events AC11-1.1 to AC13-5). Based on fig. 9, the mean
background mixing ratio of each gas is subtracted from the measured mixing ratios within the plumes. From the resulting
enhancements, the normalized excess mixing ratio $R_{XF}$ (or NEMR) is calculated according to

$$R_{XF} = \frac{\Delta[X]}{\Delta[CH_2O]} = \frac{[X_{fire}] - [X_{bkg}]}{[CH_2O_{fire}] - [CH_2O_{bkg}]} \tag{4}$$

with X being either $C_2H_2O_2$ or $C_3H_4O_2^*$. The results for each individual biomass burning event as well as the mean and range
of $R_{GF}$ and $R_{MF}$ are given in Table 3.

### 5.4.1 $R_{GF}$

$R_{GF}$ ranges between 0.024 and 0.112, with an average of $\overline{R}_{GF} = 0.065 \pm 0.048$. Repeated measurements of the same biomass
burning plumes all agree well within the error, indicating the respective plumes were in fact the major emission sources
throughout the measurements. The plume AC11-3 (fig. 7) yields the highest $R_{GF}$ of 0.097 and 0.112. Unfortunately, the plume
was not reported in the MODIS fire database. Therefore, we do not know the precise location of the fire and cannot discuss the
results with respect to the distance of the aircraft from the plume. The smallest $R_{GF}$ of 0.024 is found during a measurement
with an approximately 1000 m higher flight altitude than during the rest of the events.

Our results are consistent with recent laboratory studies on biomass burning emissions by Zarzana et al. (2018) as well as
different field measurements and satellite observations by Chan Miller et al. (2014, 2017); DiGangi et al. (2012); Wittrock
et al. (2006) and Zarzana et al. (2017). From laboratory measurements, Zarzana et al. (2018) reported an average $R_{GF}$ of
0.068 for fresh emissions and different kinds of fuels. Without further knowledge on the fuels burned during the observed
events, we cannot make firm conclusions on the photochemistry within the fire plumes. While different fuel types may lead to
changing emission factors of glyoxal, Zarzana et al. (2018) observed only little variance in $R_{GF}$ for different burning species.
The correlation of $C_2H_2O_2$ and $CH_2O$ seems to be consistent for different kinds of fuel. The lack of knowledge of the exact





**Table 3.** Inferred normalised excess mixing ratios and emission factors for individual biomass burning events (range and mean). $R_{GF}$ and $R_{MF}$ are given in $\mathrm{mole\,mole^{-1}}$. The emission factors $EF_{G}$ and $EF_{M}$ are in units of $\mathrm{g\,kg^{-1}}$.

| event | time interval [UTC] | location | altitude [km] | $R_{GF}$ | $R_{MF}$ | $EF_{G}$ | $EF_{M}$ |
|---|---|---|---|---|---|---|---|
| AC11-1.1 | 15:03:19–15:03:51 | 2.9° S, 60.1° W | 0.624–0.632 | $0.05 \pm 0.03$ | $0.09 \pm 0.13$ | $0.23 \pm 0.13$ | $0.50 \pm 0.76$ |
| AC11-1.2 | 15:03:48–15:04:16 | 2.9° S, 60.1° W | 0.615–0.647 | $0.03 \pm 0.02$ | $0.11 \pm 0.16$ | $0.16 \pm 0.12$ | $0.65 \pm 0.94$ |
| AC11-2.1 | 17:21:06–17:23:31 | 3.6° S, 60.3° W | 0.455–0.471 | $0.06 \pm 0.04$ | $0.86 \pm 0.39$ | $0.26 \pm 0.19$ | $4.96 \pm 2.6$ |
| AC11-2.2 | 17:23:31–17:24:48 | 3.6° S, 60.3° W | 0.453–0.470 | $0.06 \pm 0.04$ | $0.66 \pm 0.31$ | $0.26 \pm 0.18$ | $3.8 \pm 2.04$ |
| AC11-2.3 | 17:24:50–17:26:30 | 3.6° S, 60.3° W | 0.452–0.470 | $0.03 \pm 0.03$ | $0.51 \pm 0.26$ | $0.16 \pm 0.15$ | $2.93 \pm 1.68$ |
| AC11-3.1 | 17:43:27–17:44:41 | 2.9° S, 60.5° W | 0.450–0.456 | $0.11 \pm 0.10$ | $1.50 \pm 0.60$ | $0.52 \pm 0.47$ | $8.64 \pm 4.13$ |
| AC11-3.2 | 17:44:43–17:45:52 | 2.8° S, 60.5° W | 0.449–0.454 | $0.10 \pm 0.11$ | $1.30 \pm 0.70$ | $0.45 \pm 0.52$ | $7.49 \pm 4.49$ |
| AC13-1 | 16:33:41–16:34:20 | 11.0° S, 56.2° W | 0.923–0.934 | $0.04 \pm 0.05$ | $1.20 \pm 0.60$ | $0.18 \pm 0.23$ | $6.91 \pm 3.90$ |
| AC13-2 | 16:34:20–16:34:56 | 11.1° S, 56.2° W | 0.921–0.932 | $0.08 \pm 0.07$ | $1.40 \pm 0.70$ | $0.37 \pm 0.34$ | $8.06 \pm 4.55$ |
| AC13-3 | 16:35:39–16:36:18 | 11.2° S, 56.2° W | 0.918–0.929 | $0.03 \pm 0.04$ | $0.76 \pm 0.43$ | $0.13 \pm 0.17$ | $4.35 \pm 2.73$ |
| AC13-4 | 16:37:01–16:37:43 | 11.2° S, 56.2° W | 0.915–0.932 | $0.05 \pm 0.04$ | $0.81 \pm 0.43$ | $0.23 \pm 0.19$ | $4.67 \pm 2.76$ |
| AC13-5 | 16:52:15–16:53:03 | 11.2° S, 56.2° W | 1.849–1.191 | $0.02 \pm 0.02$ | $0.56 \pm 0.27$ | $0.11 \pm 0.10$ | $3.25 \pm 1.77$ |
| Range | | | | 0.02–0.11 | 0.09–1.50 | 0.11–0.52 | 0.50–8.64 |
| Mean | | | | $0.07 \pm 0.05$ | $0.98 \pm 0.42$ | $0.25 \pm 0.23$ | $4.70 \pm 2.70$ |

fuels burned in each fire is therefore not an issue when comparing $R_{GF}$ of the different plumes and intercepts. In two older studies, Hays et al. (2002) and McDonald et al. (2000b) report much higher $R_{GF}$ on the order of approximately 2.5 to 3 for Palmae and Poaceae (Hays et al., 2002) and even up to 4 (McDonald et al., 2000b). Our inferred $R_{GF}$ are at least one order of magnitude lower. In fact, results larger than 1 are only observed for the background-uncorrected $R_{GF}^{*}$ in the free troposphere.

During recent field measurements over the US, Zarzana et al. (2017) report $R_{GF}$ in the range 0.008 to 0.110. Due to the
very nature of airborne plume measurements, we do not know by how much the measurements differ with respect to plume age, air mass origin, atmospheric composition, measurement distance to the plume, etc. Despite the very different atmospheric backgrounds and conditions, our measurements over the rain forest agree well with their findings.

Measurements of an aged biomass burning plume yield slightly smaller $R_{GF}$ of 0.02 to 0.03 (DiGangi et al., 2012). Of all events, event AC13-5 is most likely the oldest plume measured. This is a consequence of the high flight altitude in combination
with the distance from the aircraft to the plume during the measurement. AC13-5 has the smallest $R_{GF}$ of only 0.024, which agrees with the findings of DiGangi et al. (2012). We cannot state by how much this plume measurement is influenced by the general atmospheric background. The air masses in the boundary layer are generally highly polluted with biomass burning emission of different ages. We do not know how much these differently aged emissions influenced our individual plume measurements. Therefore, we cannot draw a distinct relationship between $R_{GF}$ and the plume ages, which, due to the well-
mixed condition of the boundary layer most probably varied between several minutes to hours during all measurements.





### 5.4.2 $R_{MF}$

$R_{MF}$ ranges between 0.09 and 1.50, with an average of $\overline{R}_{MF} = (0.98 \pm 0.42)$. Three of the eight probed plumes yield $R_{MF} > 1$ (both measurements of AC11-3, AC13-1, and AC13-2). All the other plumes yield $R_{MF} < 1$, in the range of 0.09 to 0.86. For event AC11-1, $R_{GF}$ is comparable to the other results, while $R_{MF}$ is significantly smaller than the average $\overline{R}_{MF}$ by approximately a factor of 10. Based on the video imagery, the size of the observed plumes as well as the estimated distance to the aircraft are comparable during events AC11-1 and AC11-3 (fig. 7). Still, they yield very different normalised excess mixing ratios. In fact, the largest $R_{MF}$ and $R_{GF}$ are obtained during event AC11-3. Accordingly, the emission of $C_2H_2O_2$ and $C_3H_4O_2^*$ with respect to $CH_2O$ is the highest within this biomass burning plume. Interestingly, this is not connected to the size and quantity of the biomass burning events seen by the telescopes. The highest emissions (i.e. the largest plumes) were observed during event AC11-2, where three different large fires were simultaneously detected in a generally hazy atmosphere rich in fresh emissions (fig. 7). These plumes were also recorded by MCD14 on MODIS (fig. 6, red dots in the field of view of event 2.2). Laboratory measurements report $R_{MF}$ approximately between 1 and 2 for different kinds of fuels (Hays et al., 2002). This is the upper range of our observations. Instead of $R_{MF}$, Zarzana et al. (2018) report the ratio $C_3H_4O_2$ to $C_2H_2O_2$ ($R_{MG}$) to be on the order of 1.7–2.5 for burning rice straw and different kinds of canopy. Our measurements yield $R_{MG}$ in the range 1–3, with the exception of event AC11-1, where $R_{MG} < 1$ due to the small $C_3H_4O_2^*$ mixing ratios. Event AC11-3 yields $R_{MG}=1$ during both measurements. This reflects the correlated strong enhancement of both gases within the biomass burning plumes.

### 5.4.3 Biomass burning emission factors

According to Andreae (2019), the biomass burning emission factors for $C_2H_2O_2$ and $C_3H_4O_2^*$ are defined by the normalised excess mixing ratio $R_{XF}$ as

$$EF_X = R_{XF} \frac{MW_X}{MW_{CH_2O}} EF_{CH_2O} \tag{5}$$

using the molecular weight MW of each species, and the mean emission factor $EF_{CH_2O}$ of the reference trace gas $CH_2O$. As the latter was not measured during the campaign, we use the recent comprehensive compilation by Andreae (2019), which reports $EF_{CH_2O}$ with respect to different combustion processes and fuel types. For tropical forest fires, Andreae (2019) inferred mean $EF_{CH_2O}=2.4 \pm 0.63$. The resulting $EF_G$ and $EF_M$ are listed in Table 3 for each biomass burning plume intercept in units of g (target species) $kg^{-1}$ (fuel).

$EF_G$ ranges from 0.11 to 0.52 g glyoxal per $kg$ of fuel burned with a mean emission factor of $(0.25 \pm 0.23)$ $g\,kg^{-1}$. According to its high $R_{GF}$, event AC11-3 yields the highest emission factors of 0.52 and 0.45 $g\,kg^{-1}$. Five out of the twelve plume encounters yield emission factors of less than 0.2 $g\,kg^{-1}$. Corresponding to the largest $EF_G$, Andreae (2019) estimated $EF_G=0.6$ $g\,kg^{-1}$ for tropical forest fires. In the same study, glyoxal emissions from open burns of agricultural residues were estimated to be approximately 60 % lower with $EF_G=0.23$ $g\,kg^{-1}$. All biomass burning events encountered during flight AC13 were located between 11.0–11.2° S and 56.2° W. While other parts of the measurement flight were located over rain forest,





this region is largely dominated by agricultural activities. $EF_G$ of events AC13-1 to AC13-5 ranges from 0.11 to 0.37 $g\,kg^{-1}$.
Assuming that the biomass burning plumes were dominated by agricultural residues during these measurements, we infer

a corresponding mean emission factor $EF_G=0.2\,g$ glyoxal per $kg$ of open burns of agricultural residues. The range of $EF_G$
further agrees with laboratory measurements by Zarzana et al. (2018), who reported $EF_G$ in the range of 0.06 to 0.55 $g\,kg^{-1}$
depending on the fuel type.

    $EF_M$ ranges between 0.5 and 8.64 $g$ methylglyoxal per $kg$ of fuel burned, with a mean emission factor of $(4.68 \pm 2.7)\,g\,kg^{-1}$.
For each biomass burning event, $EF_M$ is at least twice as high as $EF_G$. The largest $EF_M$ were found during flight AC13, i.e.

for fires dominated by agricultural residues. Even after applaying the correction factor of 2 on the inferred $EF_M$, the results are
significantly larger than reported by Zarzana et al. (2018) from laboratory measurements, who found maximal emission factors
of approximately 2 $g\,kg^{-1}$ for burns of duff. While their absolute $EF_M$ are much lower, their results were equally variable with
$EF_M$ ranging from 0.11 to 2.0 $g\,kg^{-1}$.

## 6   Conclusions

We report on the first simultaneous measurements of $CH_2O$, $C_2H_2O_2$ and $C_3H_4O_2^*$ over remote and polluted sections of
the Amazon. The measurements were performed from on board the DLR HALO aircraft during the ACRIDICON-CHUVA
campaign in fall 2014 (Wendisch et al., 2016). The observations took place in the troposphere between 500 $m$ and 15 $km$
altitude, where air masses of different compositions were encountered. These include continental background as well as aged
and fresh biomass burning plumes.

The observed formaldehyde mixing ratios in the lower and free troposphere range from those previously reported over the
remote Pacific (Peters et al., 2012) up to those measured over continental North and South America (Kaiser et al., 2015). The
lower quartile of the measured mixing ratios appears to result from the oxidation of VOCs (mostly isoprene). In the lower
troposphere, enhanced formaldehyde mixing ratios have two major contributions: direct emission from biogenic sources or
biomass burning and secondary formation during the degradation of short-lived VOCs. In the middle and upper troposphere,

formaldehyde seems to have more constant sources and sinks than in the lower troposphere. While the sink (mostly photolysis)
may not vary significantly with altitude, the degradation of longer lived VOCs provides a comparably constant source. When
integrating our profiles, our observations are in good agreement with previously inferred formaldehyde VCDs from satellites
(De Smedt et al., 2008b; Wittrock et al., 2006; Vrekoussis et al., 2010; De Smedt et al., 2015).

    A good agreement with previous satellite measurements is also found for the lower and upper range of profile integrated

glyoxal mixing ratios (Wittrock et al., 2006; Vrekoussis et al., 2009, 2010; Alvarado et al., 2014). Analogously to the presence
of formaldehyde, the lower bound of glyoxal is due to the oxidation of precursor VOCs. Besides direct emissions from biomass
burning and secondarily formed glyoxal from short lived precursors, the oxidation of longer lived precursors like isoprene or
acetone contributes to elevated glyoxal mixing ratios in the free troposphere.

    In agreement with the study of Zarzana et al. (2017), $C_3H_4O_2^*$ mixing ratios generally exceed those of glyoxal by at least a

factor of 5 during the majority of the measurements. Yet, the shapes of the upper and lower bounds appear similar to the profile





shape of glyoxal in contrast to formaldehyde, which appears to decline faster above the boundary layer in the low-$NO_x$, high-VOC, and low-$HO_x$ environment probed during this study. The compact clustering as well as the smooth decrease of the lower quartile mixing ratios with altitude indicate the type and sink of the precursor molecules. Apparently, these precursors should have lifetimes of at least several days, and should either be constantly emitted by the biosphere (e.g. isoprene and acetone) or

occasionally during biomass burning (e.g. acetylene) before being vertically transported. $R_{GF}^*$ and $R_{MF}^*$ are found to increase with altitude due to the more rapid decrease of formaldehyde relative to glyoxal and methylglyoxal mixing ratios. Again, this behaviour expresses the different and changing fractions of the short and long-lived precursors relevant for the formation of formaldehyde as compared to those of glyoxal and methylglyoxal. In particular, the increasing $R_{GF}$ and $R_{MF}$ in the middle and upper troposphere require significant sources for both gases which are supposed to result from the oxidation of longer-lived

VOCs (Schulz et al., 2018). In fact, since both gases are known to contribute to SOA formation, the elevated mixing ratios of glyoxal and methylglyoxal in the upper troposphere further lend support to the proposed SOA formation from products of isoprene oxidation as observed by Schulz et al. (2018); Andreae et al. (2018) and Williamson et al. (2019) over the Amazon Basin and generally in the tropics.

Our inferred $R_{GF}$ in the range of 2.4–11.2 ($\overline{R}_{GF}=6.5$) for the observed biomass burning events agree well with previous

studies, both in the field over North America (Zarzana et al., 2017) as well as in the laboratory fuel study (Zarzana et al., 2018). It is further in agreement with $R_{GF}$ inferred from satellite measurements (Chan Miller et al., 2014, 2017; DiGangi et al., 2012; Wittrock et al., 2006). For $R_{MF}$, we infer a range of 0.09 to 1.5 with $\overline{R}_{MF}=0.98$ (or respectively smaller results when correcting the $C_3H_4O_2^*$ mixing ratios by a factor of $2.0\pm0.5$), which overlaps with the lower end of the range of normalized excess mixing ratios previously reported by Hays et al. (2002) for different types of fuels in the laboratory. The presented study

as well as the measurements by Hays et al. (2002) indicate a slightly larger $R_{MF}$ than estimated from $R_{MG}$ and $R_{GF}$ (appr. 0.1–0.5) reported by Zarzana et al. (2018), even though this difference may in part be a result of additional dicarbonyls included in our retrieval. In fact, when correcting $R_{MF}$ by a factor of 2, our results agree much better to the measurements by Zarzana et al. (2018). Based on $R_{GF}$ and $R_{MF}$ and average emission factors for formaldehyde estimated by Andreae (2019), we infer emission factors in the range of 0.11–0.52 ($\overline{EF}_G=(0.25\pm0.23)$ g kg$^{-1}$) for glyoxal and 0.5–8.6 ($\overline{EF}_M=(4.7\pm2.7)$ g kg$^{-1}$)

for methylglyoxal for the probed biomass burning events over the Amazon. Our inferred $EF_G$ agrees well with the range reported from laboratory measurements (Zarzana et al., 2018). Notably, our $EF_M$ is significantly larger than found in their study, also when correcting the results for possibly included additional dicarbonyls.

The presented study may be complemented by future studies with data collected during several past HALO missions, i.e. ESMVal (fall 2012), OMO (over the Mediteranean in summer 2014), EMeRGe (Europe in summer 2017 and Eastern Asia in

spring 2018), CAFE (over West Africa/tropical Atlantic in summer 2018), and Southtrac (Southern Argentina and northern Antarctica in fall 2019), which may provide further information on the role of carbonyls in global atmospheric chemistry.

*Data availability.* All the data are archived in the HALO data depository (https://halo-db.pa.op.dlr.de/mission/5), datasets 6495-6502 (mini-DOAS data), and 3440-3442 (AMTEX CO data). The data are accessible upon signing a data protocol.



*Author contributions.* TH, MK and KP operated the mini-DOAS instrument and ML and HS performed the AMTEX measurements during the ACRIDICON-CHUVA campaign. TH wrote the data retrieval software. FK performed the data analysis and wrote the paper with contributions from ML, KP and HS.

*Competing interests.* The authors declare that they have no conflict of interest.

*Financial support.* The study was funded by the German Research Foundation (DFG, HALO-SPP 1294). The contributions of FK, TH and KP were supported via the German Research Foundation (DFG) through grants PF-384/7-1, PF384/9-1, PF-384/16-1, PF-384/17, and PF-384/19. HS and ML gratefully acknowledge the support given by the DFG projects SCHL 1857/1-2, SCHL 1857/2-2, and SCHL 1857/4-1.

*Acknowledgements.* The authors gratefully acknowledge the NOAA Air Resources Laboratory (ARL) for the provision of the HYSPLIT transport and dispersion model and READY website (https://www.ready.noaa.gov) used in this publication. We acknowledge the use of imagery from the NASA Worldview application (https://worldview.earthdata.nasa.gov/), part of the NASA Earth Observing System Data and Information System (EOSDIS) and the use of data products and imagery from the Land, Atmosphere Near real-time Capability for EOS (LANCE) system operated by NASA's Earth Science Data and Information System (ESDIS) with funding provided by NASA Headquarters. We thank the Deutsches Zentrum für Luft- und Raumfahrt (DLR) for the support during the certification process of the mini-DOAS instrument and the DLR Flugexperimente Team at Oberpfaffenhofen, in particular, Frank Probst, Martina Hierle, Andreas Minikin and Andrea Hausold, for the support given during the ACRIDICON-CHUVA mission. We are grateful to Manfred Wendisch (University of Leipzig, Germany) for coordinating the campaign and to Steven Brown (NOAA Earth System Research Laboratory) for his helpful discussion of our measurements and conclusions.



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
