# Peer review of "Profiling of formaldehyde, glyoxal, methylglyoxal, and CO over the Amazon: Normalized excess mixing ratios and related emission factors in biomass burning plumes"

_Atmospheric Chemistry and Physics, 2020_

## Referee Comment (RC1) · Anonymous Referee #2 · 6 Jun 2020

The authors report results from a recent aircraft campaign over the Amazon and nearby city of Manaus. Using a mini-DOAS instrument, they measured formaldehyde, glyoxal, and methylglyoxal vertical profiles and emission ratios of glyoxal and methylglyoxal relative to formaldehyde in biomass burning plumes. They compare their results with others in the literature and find them to be well in agreement with previous studies. They also compare the total column densities to those measured by various satellites and also find good agreement.

The paper is well written and clear, the results are robust and will be a useful contribution to the fields of remote sensing and VOC oxidation in biomass burning plumes. I recommend publication, with only a few minor comments.

Line 10: The authors have not yet introduced RGF or RMF, and unless readers are already familiar with the chemical formulas for glyoxal and methylglyoxal, this may be confusing. I suggest either noting the names of the species in Line 1, or defining RGF and RMF here.

Line 29: The authors state that these compounds are emitted, but much of the paper discusses their formation mechanism, so I would change "emitted" to "emitted or formed"

Line 52: The line "glyoxal and methylglyoxal are formed by 47 % and 79 %, respectively" is slightly confusing. Does that mean 47% of the glyoxal is formed from isoprene? Or 47% of the isoprene that reacts forms glyoxal?

Line 57: Rather than citing "GEOS-Chem model simulations" for the lifetime, I would cite the papers that get that figure, such as Fu 2008

Line 93: No comma needed after "They concluded"

Line 97: If the authors want to mention the A-train, include a brief description of what that is. Otherwise, it does not seem necessary to include that information.

Line 112: German should be capitalized

Table 1: "Temperature" is misspelled

Line 172 or later: It's not clear how the [C2H2O2] and [C3H4O2*] concentrations are derived from the mini-DOAS remote sensing measurement. Are they derived from the ODlimb line-of-sight measurement? Do we interpret that as the instantaneous concentration at the altitude of the aircraft? Particularly later, when the total column measurements are compared to the satellite measurements, it would be helpful to have that distinction made.

Line 175-181: Use the terms ODlimb and ODms instead of b and c for more clarity.

Figures 2 and 3: The various shades of blue are hard to differentiate, and the markers are too small to really see the shapes.

Figure 5: What time are the MODIS satellite images from? Beginning of flight? The panel numbers a-d are missing

Figure 9: It is very difficult to see the "C-shape" curve in the CO data that the authors discuss. Consider putting the x-axis on a log scale.

Line 376: To remain consistent with the rest of the paper, use ppb instead of nmol/mol

Line 357: "leading to their generally correlated vertical profiles". Later on, the authors state that the formaldehyde vertical profiles is markedly different from the other two. Clarify this?

Figure 11: In the x-axis label, put which satellite(s) were used in each of the other papers.

Line 420: Are the emission ratios from the column integrated value or the concentration value?

Line 440: "Emissions of monoterpenes, like alpha-pinene, on the other hand, are at least one order of magnitude smaller". This statement is unclear. Smaller than what?

---

## Referee Comment (RC2) · Anonymous Referee #3 · 6 Jun 2020

This manuscript presents vertical profile measurements of formaldehyde, glyoxal, methylglyoxal, and O over the AMAzon during the HALO campaign (fall 2014) using mini-DOAS. The authors found enhanced concentrations of all four pollutants in air masses affected by biomass burning. They further calculated the normalized excess mixing ratios and relative emission factors of glyoxal and methylglyoxal from biomass burning. The normalized excess mixing ratio for glyoxal was in good agreement with other recent reports, but the value for methylglyoxal was variable and much larger than previous reports. Both of these values can be used in models to help interpret the

sources of glyoxal and methylglyoxal, which in turn helps with the analyses of satellite-based glyoxal measurements and the sources of secondary organic aerosols.

The thesis of the paper is very much of interest to the community. In addition to the well-documented measurements, I was particular interested to see the very thorough uncertainty analysis (Section 2.1) and the difference between the 'gross mixing ratio' and the 'normalized excess mixing ratios'. The paper is generally well written, clear, and quantitative, with no major flaws. There are some minor grammatic, spelling, and citation errors, but those are easy to fix. In all, I recommend publication of this paper after a minor revision.

Comments:

Lines 172-182: The writing in this paragraph is somewhat confusing. The statement of UV/vis limb measurements dividing the atmosphere in to three parts (a, b, and c) applies not only to O4 absorption but also to the absorption of the targeted species. But here the authors only discussed the implication for O4 absorption (i.e., b+c dominates, which is also true for the targeted species here). What are the implications for the retrievals of the targeted species?

Lines 194-197: "From the above discussion, .... volume.": Can the authors say something about the estimated size and orientation of this averaging air volumne?

Other minor comments:

Line 13: 'applaying' should be 'applying'

Line 31: Extra 'is' after C3H4O2*. Please remove

Line 52-69: Missing key reference for the global budget of glyoxal: Myriokefalitakis et al. (2008) Myriokefalitakis, S., M. Vrekoussis, K. Tsigaridis, F. Wittrock, A. Richter, C. Brühl, R. Volkamer, J.P. Burrows, and M. Kanakidou, 2008: The influence of natural and anthropogenic secondary sources on the glyoxal global distribution. Atmos. Chem. Phys., 8, 4965-4981, doi:10.5194/acp-8-4965-2008.

Lines 85-87: "In comparison to satellite ...biogenic precursors." This statement is missing references.

Line 241: Remove 'either of'

LIne 250: "several 100 m" should be "several hundred meters"

Figure 8 caption: "The colour coding in panels (e)), and (g)" should be "(e) and (f)"

Line 368: Extra ")" after "(Fu et al., 2008). Also, there was no budget analysis for formaldehyde in Fu et al. (2008). Please cite a relevant reference.

Line 733: Capitalize 'C' in Nature 'communications'.

---

## Referee Comment (RC3) · Anonymous Referee #1 · 9 Jun 2020

This manuscript presents DOAS measurements with an airborne instrument of formaldehyde, glyoxal and effective methylglyoxal atmospheric concentrations as well as in situ CO observations over Amazonia during a series of flights carried within the ACRIDICON-CHUVA campaign. Those measurements have been done within different type of air masses and at different altitudes: pristine tropospheric air, air masses dominated by VOC emitted of biogenic origin or affected by biomass burning events and/or pollution coming from the city of Manau. Those measurements allow deriving for those species vertical columns but more interestingly concentration profiles. The

results are extensively discussed and compared with existing data in the current literature. This work fits well in ACP and definitely deserves publication after the following minor comments are taken into account.

**Comments:**

- Having this gap within the glyoxal fit window to avoid interferences with H2O is interesting. Would it be possible to better illustrate the impact of doing this (e.g compare SCD with/without this gap)? Why is not needed for the methylglyoxal fits?

- In Fig. 8, there are some peaks in CO measurements to which no specific event is allocated and the other way round, there are biomass burning events without any increase in CO concentrations. Do we understand that? A similar comment can be done for other species. For example in the fire event 2, a spike is visible in HCHO measurements but not in C2H2O2 nor in C3H4O2, while in the event 6, this is the reverse. Are those fires from different origins?

- In figure 9, in case the measurement sampling allows to do so, it would be very interesting to see mean concentration profiles classified as a function of the type of air mass (pristine, biogenic, fires, pollution).

- During the discussion on the Rgf and Rmf profiles, I would be much more cautious as the level of noise is very high. I think that the only thing which can be said is that the ratio is lower in the boundary layer compared to higher altitudes. Other conclusions on the profile shape are quite hazardous.

- When computing Rxf, how do you define the background? Out of the different marked events (red, green, blue events), there is still a very large variability in the measured concentrations with values sometimes as large as during the selected events. So the notion of background is unclear here. Please clarify this aspect.

**Minor/Technical comments:**

- Line 13: Correct "applaying" by "applying"

- Line 31: remove "is"

- Table 1 caption: write "Temperature" instead of "Temperatur"

- Line 162: add "transfer" after "radiative"

- Line 177: replace "reminder" by "remainder"

- Line 185: the statement "dominates with > 50%" is not clear. Please rephrase.

- Figures 2 and 3: All the different blueish curves tend to mix together. Please use more contrasted colors to improve visibility.

- Lines 196-198: remove "on spatial scales…analysis of our data", which is unclear.

- Line 203: Uncapitalize "Fit"

- Line 247: add "s" to "flight" ("flights")

- Line 268 : remove "in"

- Line 269: replace "more closely" by "further"

- Figure 8: add a colorbar for Rgf/Rmf. Please mention in the caption what is the shaded blue area in the three upper panels.

- Line 313: remove either "are" or "appear"

- Line 335: this factor 2 is likely a very rough way to get C3H4O2 concentrations since I suppose that the relative amounts of the C3H4O2* family members strongly depend on the precursor concentrations, and thus on the geolocation, altitude, geophysical regime. Please add a small statement to draw attention on this.

- Line 381 and figure 9 has as consequence that the basic assumption for error estimates is not verified (line 194). I agree however that some approximations are needed for such estimates. Please state when you make this hypothesis that it is not fully correct.

- Section 5.3.1: Please mention the type of measurements that Kaiser et al. and MacDonald et al. have performed (altitude range, season, measurement type).

- Line 444: replace "Contrary" before "On contrary"

- Line 481: The sentence "As a result,... burning plumes" is a repetition of the previous one to me. I suggest you delete it.

- Line 570: replace "applaying" by "applying"

- Line 583: to my knowledge, there is very little direct emissions of HCHO from biogenic sources. It comes mostly from indirect production from other biogenically emitted VOCs such as isoprene.

---

## Author Response (AR1)

Authors point-by-point response to the referee comments RC1, RC2, and RC3 on "Profiling of formaldehyde, glyoxal, methylglyoxal, and CO over the Amazon: Normalised excess mixing ratios and related emission factors in biomass burning plumes" by Flora Kluge et al.

We are very grateful to the reviewers for there comments and overall very positive assessments of our manuscript. In the following point-by-point response, the reviewers comments are written in **bold**, our responses are marked with AC (authors comments).

The document is structured as follows:
Pages 1-5: Reviewer Comment RC1
Page 6-9: Reviewer Comment RC2
Page 10-12: Reviewer Comment 12
Page 13-14: Additional changes in the manuscript by the authors

**Reviewer Comment RC1:**

Comments:

**Having this gap within the glyoxal fit window to avoid cross interferences with H2O is interesting. Would it be possible to better illustrate the impact of doing this (e.g. compare SCD with/without gap)? Why is not needed for the methylglyoxal fits?**
AC: In fact, we tested various fitting scenarios including/excluding the inferring water vapour absorption. When including it, the residual structure was dominated by the residual water vapour absorption, casting doubts on the fitting results for glyoxal. Accordingly, we discarded the retrieval including the water vapour absorption and decided for the spectral retrieval as shown in the manuscript. The same was not the case regarding the DOAS retrieval of methylglyoxal.

**In Fig. 8, there are some peaks in CO measurements to which no specific event is allocated and the other way around, there are biomass burning events without any increase in CO concentrations. Do we understand that? A similar comment can be done for other species. For example, in the fire event 2, a spike is visible in $CH_2O$ measurements but not in $C_2H_2O_2$ nor in $C_3H_4O_2$, while in the event 6, this is the reverse. Are those fires from different origins?**
AC: In lines 208 – 214 and 294-296, we argue that a direct comparison of in-situ and remotely sensed parameters is only useful on spatial scales larger than the spatial resolution of the remotely sensed parameters. Therefore, a missing one to one correlation (c.f. at 17:30 and at 18:15 for the HALO flight AC 11, fig. 8) between the various in situ and remotely sensed parameters is per se not

astonishing. Also, $CH_2O$ absorption was measured in the UV spectral region, while $C_2H_2O_2$ and $C_3H_4O_2^*$ were measured in the vis, using two different spectrometers which are not temporally aligned. Therefore, the differences in the temporal recording time of both spectrometers may cause different detection sensitivities for specific fire events. For example, the temporal resolution of the UV spectrometer was roughly four times lower than the temporal resolution of the vis spectrometer for event 2, during flight AC11 (fig.8). While it is possible that the in situ and remote instruments probed different airmasses, this is not possible for our remote sensing measurements, since the mini DOAS telescopes all point into the same direction (with the same field of view).

**In figure 9, in case the measurement sampling allows to do so, it would be very interesting to see mean concentration profiles classified as a function of the type of air mass (pristine, biogenic, fires, pollution).**
AC: This would indeed be interesting, but how to distinguish among the four situations without any further information. From the measured data the only criteria to discriminate among the different air masses could come from the measured concentrations and possibly air mass trajectories. We used both approaches in our study, but they may at best allow us to distinguish between pristine and biogenic emission affected air masses (i.e. the data belonging to the lower quartile) on one hand and on the other hand those affected by biomass burning and air pollution (all other elevated concentrations). However, we strongly feel that such a qualitative sorting may not provide robust information on the origin of the air masses and in consequence we refrain from it.

**During the discussion on the Rgf and Rmf profiles, I would be much more cautious as the level of noise is very high. I think that the only thing which can be said is that the ratio is lower in the boundary layer compared to higher altitudes. Other conclusions on the profile shape are quite hazardous.**
AC: The larger scatter of data in particular at larger altitudes when the concentrations are low may partly or even mostly be due to the detection limits of the respective gases (see fig. 4) and thus not express atmospheric variability. Therefore, we mostly concentrate our discussion on comparisons to other studies made at low altitudes. The paragraph in lines 494-506 has been rephrased to emphasize the high level of noise:
"Finally, the vertical profile of $R_{GF}^*$ indicates slightly elevated ratios above the boundary layer and in the free troposphere (fig. 12, panel b). Within the boundary layer, $R_{GF}^*$ remains approximately constant. Both features were previously observed by Kaiser et al., (2015) for altitudes up to 6 km, however less pronounced. The increase in $R_{GF}^*$ between 2-10 km appears most pronounced just above the boundary layer (at about 2 km), where $CH_2O$ and $C_2H_2O_2$ mixing ratios are still significantly above the detection limits. Notably, the correlation of $CH_2O$ and $C_2H_2O_2$ is larger within the boundary layer than in the free troposphere. When discussing the profile shape of $R_{GF}^*$, one has to keep in mind

the very low $CH_2O$ and $C_2H_2O_2$ mixing ratios in the upper troposphere and the increasing influence of measurement noise on the inferred mixing ratios. Above 6 km, we observe mean $CH_2O$ and $C_2H_2O_2$ of only $54 \pm 40$ ppt and $15 \pm 5$ ppt, respectively, and accordingly, $R^*_{GF}$ varies on average by 60% among the different measurements within the same altitude range."

**When computing Rxf, how do you define the background? Out of the different marked events (red, green, blue events), there is still a large variability in the measured concentrations with values sometimes as large as during the selected events. So the notion of background is unclear here. Please clarify this aspect.**

AC: For clarification of the background definition, lines 529-535 have been changed to:

"Due to the lack of respective $C_2H_2O_2$ and $C_3H_4O_2$ measurements, we infer mean background mixing ratios $[X]_{bkg}$ for all three gases by binning the data in 100 m altitude stacks and calculating the mean of the lower data quartile for each bin as displayed in fig. 9 (grey line). As fig. 10 shows for the case of formaldehyde, the such defined background mixing ratios approximately correspond to formaldehyde measurements of pristine air masses above the Western Pacific Ocean by Peters et al., (2012). In order to detect enhancements due to the plumes, $[X]_{bkg}$ is then subtracted from the measured mixing ratios. From the resulting enhancements, the normalized excess mixing ratio..."

Minor/Technical comments:

**Line 13: Correct "applaying" by "applying"**
AC: The typo has been corrected accordingly (line 14).

**Line 31: remove "is"**
AC: The text has been rephrased accordingly (line 32).

**Table 1 caption: write "Temperature" instead of "Temperatur"**
AC: The typo has been corrected accordingly.

**Line 162: add "transfer" after "radiative"**
AC: The text is rephrased accordingly (line 172).

**Line 177: replace "reminder" by remainder"**
AC: The typo has been corrected accordingly (line 189).

**Line 185: the statement "dominates with >50%" is not clear. Please rephrase.**
AC: The sentence in lines 197-199 has been changed to:
"Up to 10 km altitude and for both investigated wavelengths, the absorption

within the line of sight of the telescopes contributes with more than 50% to the total $O_4$ absorption. A relative minimum can be seen at the top of the planetary boundary layer..."

**Figures 2 and 3: All the different blueish curves tend to mix together. Please use more contrast colours to improve visibility.**
AC: The colors of the curves in fig. 2 and 3 have been changed accordingly.

**Lines 196-198: remove "on spatial scales...analysis of our data", which is unclear.**
AC: As outlined above, lines 208–214 and 294-296 describe why we refrain from directly comparing the in situ and remotely measured data. This seems important for the correct interpretation of fig. 8 as well as later in the manuscript, when inferring emission ratios with respect to formaldehyde instead of CO.

**Line 203: Uncapitalize "Fit"**
AC: "Fit" has been uncapitalized (line 220).

**Line 247: add "s" to "flight" ("flights")**
AC: The text has been rephrased accordingly (line 264).

**Line 268: remove "in"**
AC: "in" has been removed (line 285).

**Line 269: replace "more closely" by "further"**
AC: The text has been rephrased accordingly (line 286).

**Figure 8: add a colour bar for Rgf/Rmf. Please mention in the caption what is the shaded blue area in the upper three panels.**
AC: The color coding of $R^*_{GF}/R^*_{MF}$ has been removed as it did not provide any additional information. The sentence
"The shaded blue area in panels a to c shows the respective measurement uncertainty."
has been added to the figure caption.

**Line 313: remove either "are" or "appear"**
AC: The text has been rephrased accordingly (line 330).

**Line 335: this factor 2 is likely a very rough way to get C3H4O2 concentrations since I suppose that the relative amounts of the C3H4O2\* family members strongly depend on the precursor concentrations, and thus on the geolocation, altitude, geophysical regime. Please add a small statement to draw attention on this.**

AC: We accordingly added the following sentence in lines 355-356:
"However, we note that the factor of $2.0 \pm 0.5$ may largely depend on the precursor concentrations, and thus on the geolocation, altitude, and geophysical regime, and therefore barely provides more than a hint on its true size."

**Line 381 and figure 9 has as consequence that the basic assumption for error estimates is not verified (line 194). I agree however that some approximations are needed for such estimates. Please state when you make this hypothesis that it is not fully correct.**
AC: We agree and have accordingly changed the sentence in lines 215-216 to:
"Summing-up all described uncertainties, the precision error of the combined methods can be approximately calculated according to eq. (1) as..."

**Section 5.3.1: Please mention the type of measurements that Kaiser et al. and MacDonald et al. have performed (altitude range, season, measurement type).**
AC: The text in lines 458-462 has been changed to:
"The comparison with Kaiser et al., (2015) shows, that the inferred $R_{GF}^*$ is notably larger than during most of their in situ measurements at altitudes ranging from the ground up to 6 km over the southeastern US in June-July 2013. Our results are in much better agreement with the $R_{GF}^*$ inferred by MacDonald et al., (2012) from ground based DOAS measurements for altitudes between 0 and 1000 m over a southeast Asian tropical rainforest in April-July 2008."

**Line 444: replace "Contrary" before "On contrary"**
AC: The text has been rephrased accordingly (line 475).

**Line 481: The sentence "As a result, ... burning plumes" is a repetition of the previous one to me. I suggest you delete it.**
AC: The sentence has been deleted (line 505).

**Line 570: replace "applaying" by "applying"**
AC: The typo has been corrected accordingly (line 609).

**Line 583: to my knowledge, there is very little direct emissions of HCHO from biogenic sources. It comes mostly from indirect production from other biogenically emitted VOCs such as isoprene.**
AC: The sentence in lines 622-624 has been changed to:
"In the lower troposphere, enhanced formaldehyde mixing ratios have two major contributions: direct emission e.g. from biomass burning, and secondary formation during the degradation of short-lived VOCs, like isoprene."

**Reviewer Comment RC2:**

Comments:

**Line 10: The authors have not yet introduced RGF or RMF, and unless readers are already familiar with the chemical formulas for glyoxal and methylglyoxal, this may be confusing. I suggest either noting the names of the species in Line 1, or defining RGF and RMF here.**
AC: We agree and have changed lines 1-4 to:
"We report on airborne measurements of tropospheric mixing ratios and vertical profiles of formaldehyde ($CH_2O$), glyoxal ($C_2H_2O_2$), methylglyoxal and higher carbonyls ($C_3H_4O_2^*$) (see below), and carbon monoxide (CO) over the Amazon Basin during the ACRIDICON-CHUVA campaign from the German High Altitude and Long-range research aircraft (HALO) in fall 2014."
Additionally, in lines 11 and 13 has been added:
"The mean glyoxal to formaldehyde ratio $R_{GF}$... The mean methylglyoxal to formaldehyde ratio $R_{MF}$..."

**Line 29: The authors state that these compounds are emitted, but much of the paper discusses their formation mechanism, so I would change "emitted" to "emitted or formed".**
AC: Line 30 has been rephrased to:
"Among the manifold of species emitted in large amounts by fires or formed in their plumes are carbonyl compounds..."

**Line 52: The line "glyoxal and methylglyoxal are formed by 47% and 79%, respectively" is slightly confusing. Does this mean 47% of the glyoxal is formed from isoprene? Or 47% of the isoprene that reacts forms glyoxal?**
AC: Yes, 47% of the total atmospheric glyoxal is formed from isoprene oxidation. To clarify this, the sentence in lines 54-56 has been changed to:
"Similar to formaldehyde, large parts of the atmospheric glyoxal (47%) and methylglyoxal (79%) are thought to be formed during the oxidation of isoprene emitted by vegetation (Fu et al., 2008, Wennberg et al., 2018)."

**Line 57: Rather than citing "GEOS-Chem model simulations" for the lifetime, I would cite the papers that get that figure, such as Fu 2008**
AC: The text has been rephrased accordingly (line 60).

**Line 93: No comma needed after "They concluded"**
AC: The comma has been erased (line 99).

**Line 97: If the authors want to mention the A-train, include a brief**

description of what that is. Otherwise, it does not seem necessary to
include that information.
AC: In lines 102-103, the sentence has been changed to:
"More recently, (Stavrakou et al., 2016) examined emissions of crop residue fires
in the North China Plain using data from the OMI satellite."

**Line 112: German should be capitalized**
AC: German has been capitalized (line 118).

**Table 1: "Temperature" is misspelled**
AC: The typo has been corrected accordingly.

**Line 172 or later: It's not clear how the [C2H2O2] and [C3H4O2*]
concentrations are derived from the mini-DOAS remote sensing mea-
surement. Are they derived from the ODlimb line-of-sight measure-
ment? Do we interpret that as the instantaneous concentration at
the altitude of the aircraft? Particularly later, when the total col-
umn measurements are compared to the satellite measurements, it
would be helpful to have that distinction made.**
AC: Yes, we use the line-of-sight limb measurement of target and scaling gas
to derive mixing ratios of the target gas for the altitude range of the aircraft,
from where most of the absorption comes from. This is quantified by the so-
called $\alpha$-factors, which give the fraction of the absorption in altitude layer $z_i$
relative to the total atmospheric absorption. They are derived from our DOAS
measurement ($\mathrm{SCD}_X$) and radiative transfer modelling of the respective Box air
mass factors ($\mathrm{B}_{X_i}$):

$$\alpha_j = \frac{\left(SCD_X - \sum_{i \neq j} [X]_i \, B_{X_i} z_i\right)}{SCD_X} \tag{1}$$

To clarify this aspect, lines 159-161 have been rephrased to:
"From the mini-DOAS limb measurements of the total slant column density,
the concentration $[X]_j$ of the trace gas X in the atmospheric layer j (i.e., the
altitude of the aircraft) is then determined from..."
Additionally, we have added in lines 167-169:
"When needed, the total atmospheric column density of the respective gases is
approximated by integrating the measured lower and higher quartile profiles in
incremental steps of 100m each (fig. 9)."

**Line 175-181: Use the terms ODlimb and ODms instead of b and c
for more clarity.**
AC: The text has been rephrased accordingly (lines 190-193).

**Figures 2 and 3: The various shapes of blue are hard to differentiate,
and the markers are too small to really see the shapes.**

AC: The colors of the curves in fig. 2 and 3 have been changed accordingly.

**Figure 5: What time are the MODIS satellite images from? Beginnings of the flight? The panels numbers a-d are missing**
AC: Panel numbers have been added within the figure panels. The time of each MODIS satellite image has been added to the figure caption:
"The MODIS satellite images are from 14:10 UTC (Sept. 11, 2014), 14:30 UTC (Sept. 16, 2014), 14:20 UTC (Sept. 18, 2014), and 17:55 UTC (Sept. 19, 2014). The images were taken from NASA WORLDVIEW https://worldview.earthdata.nasa.gov/."

**Figure 9: It is very difficult to see the "C-shape" curve in the CO data that the authors discuss. Consider putting the x-axis on a log scale.**
AC: The axis scaling has been changed accordingly and we have added in the figure caption:
"Note the logarithmic scale of the x-axis in panel a."

**Line 376: To remain consistent with the rest of the paper, use ppb instead of nmol/mol**
AC: The text has been rephrased accordingly (line 401).

**Line 357: "leading to their generally correlated vertical profiles". Later on, the authors state that the formaldehyde vertical profile is markedly different from the other two. Clarify this?**
AC: For clarification, line 376-378 has been changed to:
"As their dominant source, isoprene globally accounts for 67%, 47%, and 79% of the annual sources of formaldehyde, glyoxal, and methylglyoxal (Fu et al., 2008), respectively, leading to their rapidly decreasing vertical profiles."

**Figure 11: In the x-axis label, put which satellite(s) were used in each of the other papers.**
AC: The figure has been changed accordingly.

**Line 420: Are the emission ratios from the column integrated value or the concentration value?**
AC: The emission ratios are derived from the inferred mixing ratios. For clarification, line 447 has been changed to:
"Instead, we use the inferred mixing ratios..."

**Line 440: "Emissions of monoterpenes, like alpha-pinene, on the other hand, are at least one order of magnitude smaller". This statement is unclear. Smaller than what?**
AC: This compares the emissions of monoterpenes to those of isoprene (which dominate). Accordingly, line 470 has been changed to:

"Apparently, emissions of monoterpenes, like alpha-pinene, are at least one order of magnitude smaller than isoprene emissions."

**Reviewer Comment RC3:**

Comments:

**Lines 172-182: The writing in this paragraph is somewhat confusing. The statement of UV/vis limb measurements dividing the atmosphere in to three parts (a, b, and c) applies not only to O4 absorption but also to the absorption of the targeted species. But here the authors only discussed the implication for O4 absorption (i.e., b-c dominates, which is also true for the targeted species here). What are the implications for the retrievals of the targeted species?**

AC: In order to clarify this, we added in lines 186-187 and 195-196:

"In the following, we discuss the significance of the different contributions exemplarily for $O_4$. Evidently, the same tri-partitioning applies for all other gases of interest, however differently weighted as expressed by the respective alpha-factors."

"Figure 2 illustrates the contributions to $OD_{meas}$ at 343.7 nm (panel 1a) and 477.3 nm (panel 2a) for $O_4$ as a function of the flight time for Sept. 16, 2014."

**Lines 194-197: "From the above discussion, ... volume.": Can the authors say something about the estimated size and orientation of this averaging volume?**

AC: Lines 208-212 have been changed to:

"From the above discussion, it also becomes clear that our air-borne UV/vis limb measurements average over some atmospheric volume, which is determined by the viewing angle of the telescope lenses $(0.3°)$, the light path length, and the aircraft displacement during the time of measurement, both of which are in the order of several kilometers (for details see sect. 3, and fig. 6). This large sampling volume precludes direct comparisons with in situ measured quantities on spatial scales smaller than the current averaging volume."

Other minor comments:

**Line 13: "applaying" should be "applying"**
AC: The typo has been corrected accordingly (line 14).

**Line 31: Extra "is" after C3H4O2*. Please remove.**
AC: The extra 'is' has been erased (line 32).

**Line 52-69: Missing key reference for the global budget of glyoxal: Myriokefalitakis et al. (2008) Myriokefalitakis, F., M. Vrekoussis, K. Tsigaridis, F. Wittrock, A. Richter, C. Brühl, R. Volkamer, J.P. Burrows, and M. Kanakidou, 2008: The influence of natural and anthropogenic secondary sources on the glyoxal global distribution.**

Atmos. Chem. Phys.,8,4965-4981, doi:10.5194/acp-8-4965-2008.
AC: The missing reference has been added in line 65-66:
"Satellite observations have shown strongly enhanced vertical column densities of atmospheric glyoxal above this region (e.g., Myriokefalitakis et al., 2008)."

**Lines 85-87: "In comparison to satellite. . . biogenic precursors." This statement is missing references.**
AC: The sentence has been changed in lines 90-93 to:
"In comparison to satellite measurements from SCIAMACHY and GOME-2, several studies have found evidence that the models underestimate global glyoxal emissions, when not considering additional biogenic sources (Myriokefalitakis et al., 2008, Stavrakou et al., 2009b, Lerot et al., 2010)." Accordingly, the following references have been added:
Myriokefalitakis, S., et al. "The influence of natural and anthropogenic secondary sources on the glyoxal global distribution." (2008).
Stavrakou, T., et al. "The continental source of glyoxal estimated by the synergistic use of spaceborne measurements and inverse modelling." (2009).
Lerot, C., et al. "Glyoxal vertical columns from GOME-2 backscattered light measurements and comparisons with a global model." Atmos. Chem. Phys 10.24 (2010): 12-059.

**Line 241: Remove "either of"**
AC: The text has been rephrased accordingly (line 258).

**Line 250: "several 100m" should be "several hundred meters"**
AC: The text has been rephrased accordingly (line 267).

**Figure 8 caption: "The colour coding in panels (e)), and (g)" should be "(e) and (f)"**
AC: The sentence and the color coding have been removed.

**Line 368: Extra ")" after (Fu et al., 2008). Also, there was no budget analysis for formaldehyde in Fu et al. (2008). Please cite a relevant reverence.**
AC: The additional ")" refers to line 389-393: "(mostly. . . Fu (2008))." The sentence has been changed to:
"Besides their common dominant precursor, additional sources of the gases differently influence their local distribution (mostly combustion processes and oxidation of other biogenic/anthropogenic hydrocarbons in the case of formaldehyde (Lee et al., 1998, Liu et al., 2007, Fortems et al., 2012), oxidation of acetylene in the case of glyoxal, and oxidation of acetone in the case of methylglyoxal (Fu et al., 2008)), and might therefore differently contribute to our measurements. A recent study additionally discussed the oxidation of aromatics as possible relevant source of atmospheric glyoxal and methylglyoxal (Taraborrelli

et al., 2020)."

These additional references for the budget analysis of formaldehyde are now included:

Lee, Y-N., et al. "Atmospheric chemistry and distribution of formaldehyde and several multioxygenated carbonyl compounds during the 1995 Nashville/Middle Tennessee Ozone Study." Journal of Geophysical Research: Atmospheres 103.D17 (1998): 22449-22462.

Fortems-Cheiney, A., et al. "The formaldehyde budget as seen by a global-scale multi-constraint and multi-species inversion system." Atmospheric Chemistry & Physics Discussions 12.3 (2012).

Liu, L., et al. "Photochemical modelling in the Po basin with focus on formaldehyde and ozone." (2007).

**Line 733: Capitalize "C" in Nature "communications".**
AC: Communications has been capitalized (line 768).

**Additional changes in the manuscript by the authors:**

**Title, line 10, line 517, table 3 (caption), line 575, line 589:** normalised → normalized

**Line 12:** "...chemical transport models...". has been added

**Line 44:** catalysed → catalyzed

**Line 57:** "...isoprene..." has been added.

**Line 67:** anthropogenicly → anthropogenically

**Line 69:** New reference added: (Taraborelli et al., 2020)

**Lines 91-92:** The sentence has been changed to: "...evidence that the models underestimate global glyoxal emissions, when not considering additional biogenic sources..."

**Lines 121-122:** The sentence has been changed to: "...and combined with previously reported emission factors of..."

**Line 126:** "present" has been removed

**Line 132, 232, 454, 485:** analysed → analyzed

**Figure 1 (caption, line 4):** fit → fitted

**Line 177:** A comma has been added: "...simulated, and..."

**Figure 2 (caption):** The sentence has been changed to: "The average line-of-sight photon path length...is plotted in panels 1b, 2b."

**Figure 2 (caption, line 1,4,5,6), Figure 3 (caption, line 1), Figure 8 (caption, line 1,2), Figure 9 (caption, line 1,3), lines 195, 298, 304, 308, 319, 327, 329, 358, 371, 408, 454, 495, 508:** Parentheses removed

**Figure 2, caption:** colour → color

**Line 203:** The sentence has been changed to: "...maximum values around 100 km."

**Lines 225,226:** Spacing corrected

**Line 232:** on the order → in the order

**Line 240, Figure 6 (caption), line 467:** centre → center

**Figure 6 (caption) and Figure 12 (caption):** colour coded → color-coded

**LIne 289:** Environmenral → Environmental

**Line 310:** "...passed directly." → "...directly passed."

**Line 328:** "...photochemical and..." has been added.

**Figure 9 (caption):** "The black line shows..." → "The black lines show..."

**Figure 10 (caption):** "...(black)." → "...(grey)."

**Line 344-345:** To the sentence has been added: "...or to direct formation from longer-lived VOCs (e.g. phenols) (Andreae et al., 2001; Taraborrelli et al., 2020)."

**Line 381:** lead → contribute significantly

**Line 394:** anthropogenic emitted trace → anthropogenically emitted trace gas

**Line 400:** The sentence has been changed to: "...typical for air above..."

**Line 403:** To the sentence has been added: "...or a result of longer-lived oxidized aromatics (Taraborrelli et al., 2020)."

**Line 408:** The sentence has been changed to: "...contribution of the different precursor gases, their sinks due to photolysis and reaction with..."

**Figure 11 (caption), line 422:** column amounts → column densities

**Line 424, 438:** "panel" has been added

**Line 456:** Spacing corrected

**Line 463:** "of approximately 0.35" has been removed (repetition of line 456)

**Line 512:** previously described → described above

**Line 515:** "still missing" has been replaced by "not yet available"

**Line 518:** were → was

**Line 541, 543, 545, 564:** rounding corrected to two decimals instead of three

**Line 559:** airborne → air-borne

**Line 578:** seen → monitored

**Line 621:** To the sentence has been added: "...oxidation of methane and to a lesser degree of VOCs."

**Lines 633-638:** The following sentences have been added: "Taraborrelli et al., (2020) report in a recent study on long-lived VOCs like aromatics as a potential additional source of glyoxal in the free troposphere on a regional scale. Recent findings by Alvarado et al., (2020) reveal long range transport of glyoxal and formaldehyde within biomass burning plumes reaching distances of 1500 km downwind and thus enlarging the assumed local influence of biomass burning events on the tropospheric distribution of both gases. The time needed to reach such distances is in the order of several days, which implies the formation of glyoxal and formaldehyde from longer-lived precursors within the plumes."

**Line 645:** "aromatics" has been added

**Line 647:** behaviour → behavior

[revised manuscript text omitted]
 0.2 $\mathrm{g\,kg}^{-1}$. Corresponding to the largest $\mathrm{EF_G}$, Andreae and Merlet (2001b) estimated $\mathrm{EF_G} = 0.6\ \mathrm{g\,kg}^{-1}$ for tropical forest fires. In the same study, glyoxal emissions from open burns of agricultural residues were estimated to be approximately 60 % lower with $\mathrm{EF_G} = 0.23\ \mathrm{g\,kg}^{-1}$. All biomass burning events encountered during flight AC13 were located between 11.0–11.2° S and 56.2° W. While other parts of the measurement flight were located over rain forest, this region is largely dominated by agricultural activities. $\mathrm{EF_G}$ of events AC13-1 to AC13-5 ranges from 0.11 to 0.37 $\mathrm{g\,kg}^{-1}$. Assuming that the biomass burning plumes were dominated by agricultural residues during these measurements, we infer a corresponding mean emission factor $\mathrm{EF_G} = (0.25 \pm 0.23)$ g glyoxal per $\mathrm{kg}$ of open burns of agricultural residues. The range of $\mathrm{EF_G}$ further agrees with laboratory measurements by Zarzana et al. (2018), who reported $\mathrm{EF_G}$ in the range of 0.06 to 0.55 $\mathrm{g\,kg}^{-1}$ depending on the fuel type.

[revised manuscript text omitted]

---

## Author Response (AR2)

Profiling of formaldehyde, glyoxal, methylglyoxal, and CO over the Amazon: Normalised excess mixing ratios and related emission factors in biomass burning plumes

Interactive comment, editor review

*Editor review: The reply to the first reviewer comment RC1 is lacking quantitative information. Please clarify how significant is the impact of fitting/not fitting water vapour absorption in the glyoxal retrieval scheme. E.g. give an estimate of the typical percent change of the dSCD when including/excluding H2O, for cases where the signal is strong enough. The reason why H2O misfits are problematic for glyoxal but not for methylglyoxal also remains a bit obscure.*

Our responses:

Glyoxal: For a better comparison of the glyoxal retrieval including/excluding the water absorption at 442 nm (the 7ν water vapour band), we analysed the DOAS retrieval using a fitting window ranging from 425 to 439 nm and simultaneously at 447 to 465 nm (as described in the manuscript) as well as a continuous spectral range from 425 to 465 nm. An example retrieval using the continuous spectral range of the same skylight spectrum as shown in the manuscript (fig. 1, panel 2) is attached to this response (fig. 1), and a scatter plot of the dSCD's of both retrievals is shown in fig. 2.

For altitudes below 2 km flight altitude, where the water vapour and glyoxal absorption are the strongest, the dSCD's obtained from the continuous fit window are typically 12% smaller than those discarding the 7ν water vapour absorption band. This is presumably because part of the glyoxal absorption structure between 425 to 430 nm is removed by the not well represented weak water vapour band located between the 7ν+δ and 7ν bands (c.f., see Lampel et al., Atmos. Meas. Tech., 8, 4329–4346, https://doi.org/10.5194/amt-8-4329-2015, 2015). For larger altitudes and with decreasing water vapour and glyoxal absorption, the difference may even become larger and more variable with a mean percentage change as large as 80%. The dSCD's inferred from the continuous fitting window typically remain smaller than those discarding the 7ν water vapour band as shown in the attached fig. 2.

Methylglyoxal and higher carbonyls, briefly called methylglyoxal*: The mean difference including/excluding the 7ν water vapour band is in the order of 75% for observations below 2 km and up to 145% below 6 km flight altitude, with the smaller methylglyoxal* dSCD's obtained from the continuous fitting range. An example retrieval using the fitting range excluding the water band at 442 nm is attached (fig. 3) using the same skylight spectrum as shown in fig. 1, panel 3 in the manuscript for the continuous fit range.

As for the glyoxal retrieval, the relative difference among both retrievals increases with the flight altitude, i.e. with smaller methylglyoxal* and water vapour absorption (a scatter plot corresponding to figure 2 is attached (fig. 4)). At this point, the potential causes for the interference of the 7ν water vapour band and methylglyoxal* absorption bands in the retrieval are subject to further investigation. From the spectral retrieval, we conclude that when discarding the 7ν water band, the 6ν+δ band becomes the relevant absorption band for the retrieval of water vapour, while any remaining spectral structure around the 7ν water vapour is attributed to the absorption of methylglyoxal and 2,3-butanedione (and higher carbonyls). As a result, the amplitudes for methylglyoxal and 2,3-butanedione are increasing by the given amounts (i.e. on average by 75% below 2 km and on average by 145% below 6 km), while the overall spectral residuals are decreasing. Based on fig. 4 and our flight analysis, this effect seems to be enhanced at lower flight altitudes and when passing biomass burning plumes. The latter clearly demonstrates the presence of elevated amounts of methylglyoxal and 2,3-butanedione (and higher carbonyls) in the biomass burning plumes.

However, as described in the manuscript, the spectral resolution of the spectrometers does not allow for a more selective spectral retrieval of methylglyoxal, 2,3-butanedione, and higher carbonyls, and thus we are left to decide as to whether the inclusion/exclusion of the 7v water band is more appropriate.  Mainly based on the reduced residual structure, we decided to include the 7v water vapour band in the retrieval in a similar way as Zarzana et al., (2018). We note however, that the stated uncertainty of the methylglyoxal* retrieval (a factor of 2 for altitudes <2 km and somewhat larger for altitudes <6 km) is larger than the mean difference of 75% or 145%, respectively, between both retrieval settings, and thus the uncertainties arising from the above mentioned interference are covered by our uncertainty estimate.

Attached figures:

Figure 1: Inferred absorption spectrum and residual structure of glyoxal for the measurement at 15:06 UTC during the flight on Sept. 11, 2014 (AC11) according to fig. 1, panel 2 in the manuscript. The continuous fit range from 420 - 465 nm includes the 7v water vapour band.

Figure 2: Differential slant column densities of glyoxal based on a spectral retrieval excluding the 7v water vapour band (x-axis) and including it (y-axis) from measurement flights AC09, AC11, AC12, and AC13. The colour coding shows the respective flight altitudes.

Figure 3: Inferred absorption spectrum and residual structure of methylglyoxal* for the measurement at 14:53 UTC during the flight on Sept. 11, 2014 (AC11) according to fig. 1, panel 3 in the manuscript. The discrete fit range from 420 – 439 and 447 – 475 nm includes the 7v water vapour band.

Figure 4: Differential slant column densities of methylglyoxal* based on a spectral retrieval excluding the 7v water vapour band (x-axis) and including it (y-axis) from measurement flights AC09, AC11, AC12, and AC13. The colour coding shows the respective flight altitudes.

No changes to the manuscript Version 3 from July 14[th] have been made.

[Figure]

**Figure 1**

[Figure]

**Figure 2**

[Figure]

**Figure 3**

[Figure]

**Figure 4**     fit window 420-439nm and 447-475nm